# CONTROLAR: CONTROLLABLE IMAGE GENERATION WITH AUTOREGRESSIVE MODELS

**Zongming Li**[1*]**, Tianheng Cheng**[1*]**, Shoufa Chen**[2]**, Peize Sun**[2]**, Haocheng Shen**[3]**,
Longjin Ran**[3]**, Xiaoxin Chen**[3]**, Wenyu Liu**[1] **& Xinggang Wang**[1†]
[1] School of EIC, Huazhong University of Science and Technology
[2] Department of Computer Science, The University of Hong Kong
[3] vivo AI Lab

## ABSTRACT

Autoregressive (AR) models have reformulated image generation as *next-token prediction*, demonstrating remarkable potential and emerging as strong competitors to diffusion models. However, control-to-image generation, akin to ControlNet, remains largely unexplored within AR models. Although a natural approach, inspired by advancements Large Language Models, is to tokenize control images into tokens and prefill them into the autoregressive model before decoding image tokens, it still falls short in generation quality compared to ControlNet and suffers from inefficiency. To this end, we introduce ControlAR, an efficient and effective framework for integrating spatial controls into autoregressive image generation models. Firstly, we explore control encoding for AR models and propose a lightweight control encoder to transform spatial inputs (*e.g.*, canny edges or depth maps) into control tokens. Then ControlAR exploits the *conditional decoding* method to generate the next image token conditioned on the per-token fusion between control and image tokens, similar to positional encodings. Compared to prefilling tokens, using conditional decoding significantly strengthens the control capability of AR models but also maintains the model efficiency. Furthermore, the proposed ControlAR surprisingly empowers AR models with arbitrary-resolution image generation via conditional decoding and the specific controls. Extensive experiments can demonstrate the controllability of the proposed ControlAR for the autoregressive control-to-image generation across diverse inputs, including edges, depths, and segmentation masks. Furthermore, both quantitative and qualitative results indicate that ControlAR surpasses previous state-of-the-art controllable diffusion models, *e.g.*, ControlNet++. The code, models, and demo will soon be available at https://github.com/hustvl/ControlAR.

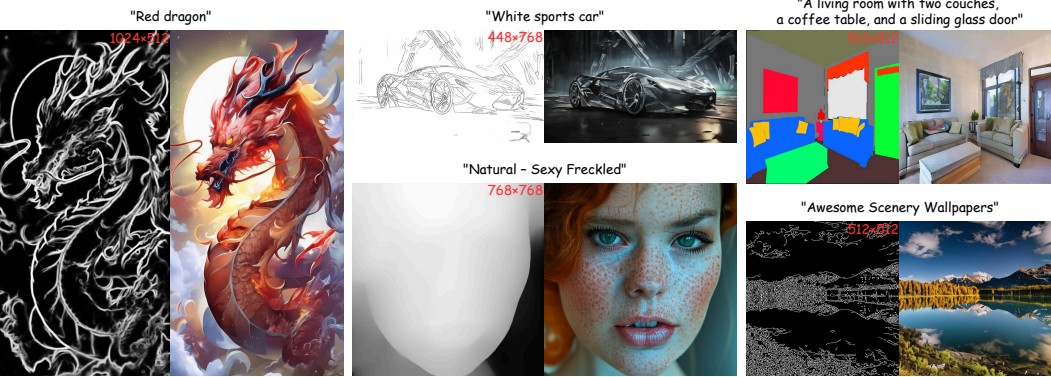

Figure 1: **Arbitrary-resolution images generated by ControlAR.** Our ControlAR extends autoregressive models, *e.g.*, LlamaGen (Sun et al., 2024), to generate high-quality images using spatial controls and expands the capability of autoregressive models to any-resolution image generation.

*Equal contributions; †Corresponding author: xgwang@hust.edu.cn.

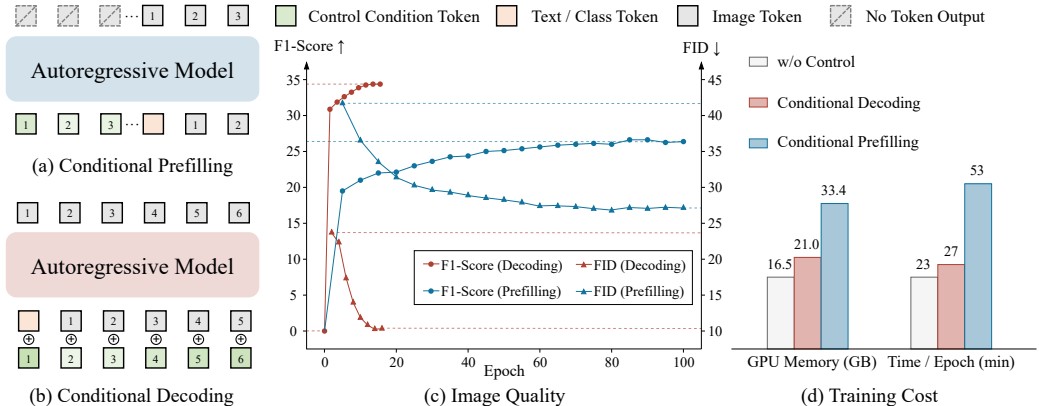

Figure 2: **Comparison between Conditional Prefilling *v.s.* Conditional Decoding.** We encode the spatial control images into a sequence of control tokens for autoregressive models. (a) Conditional Prefilling: control condition tokens are prefilled into the autoregressive model before the first image token is generated. (b) Conditional Decoding: each image token is fused with the control condition token to predict the next image token. (c) Image Quality: we compare the performance (*i.e.*, F1-Score and FID) across training epochs between conditional decoding and prefilling. It's remarkable that conditional decoding outperforms conditional prefilling in terms of performance and training convergence speed. (d) Training cost: conditional prefilling significantly increases the training memory (+59.1%) and training latency (+96.3%) compared to conditional decoding.

# 1 INTRODUCTION

Recent advancements in image generation have led to the emergence of various models that leverage text-to-image diffusion models (Saharia et al., 2022; Ho et al., 2022; Rombach et al., 2022; Podell et al., 2023) to generate high-quality visual content. Among them, several works (Zhang et al., 2023a; Li et al., 2024b; Qin et al., 2023b) such as ControlNet (Zhang et al., 2023a), have explored adding conditional controls to text-to-image diffusion models and allowed for image generation according to the precise spatial controls, *e.g.*, edges, depth maps, or segmentation masks. The control-to-image diffusion models have impressively enhanced the versatility of these models in applications ranging from creative design to augmented reality.

Despite the success of diffusion models, most recent works reveal the potential of autoregressive models for image generation, *e.g.*, LlamaGen (Sun et al., 2024) follows the architecture of Llama (Touvron et al., 2023) to achieve image generation and obtained remarkable results. Moreover, several works (Kondratyuk et al., 2024; Gao et al., 2024) have explored autoregressive models for video generation and achieved promising results, further demonstrating the great potential of autoregressive models for visual generation. However, controlling autoregressive models as a crucial direction remains unexplored, making it challenging for autoregressive models to achieve same level of fine-grained control as diffusion models. In contrast to controllable diffusion models, adding conditional controls to autoregressive models is not straightforward because of two major challenges: (1) *how to encode 2D spatial control images for autoregressive models* and (2) *how to guide image generation with encoded controls*. Specifically, diffusion models directly use the 2D features of control images and control the generated image through pixel-wise feature fusion. However, autoregressive models adopt sequence modeling and next-token prediction to perform image generation sequentially. Therefore, the techniques proposed in (Zhang et al., 2023a; Li et al., 2024b) are infeasible in autoregressive models.

In this paper, we delve into the two aforementioned challenges and introduce the **Control**lable **A**uto**R**egressive (ControlAR) framework to enhance the control capabilities of autoregressive image generation models such as LlamaGen (Sun et al., 2024) or AiM (Li et al., 2024a). Firstly, we propose a control encoder to obtain sequential encodings of control images and output the control tokens, which are more suitable than 2D control features for autoregressive models. Instead of directly replicating the modules of diffusion models for control feature extraction, we use a Vision Transformer (ViT) as the encoder and further investigate the most effective ViT pre-training scheme, *e.g.*, vanilla (Dosovitskiy, 2020) or self-supervised (Oquab et al., 2023) for encoding spatial controls towards image generation. Secondly, we naturally consider that directly prefilling control tokens,

inspired by Large Language Models and prompt techniques (Pope et al., 2023), can provide a simple and effective autoregressive control approach, as shown in Fig. 2 (a). However, it struggles to achieve satisfactory results, *i.e.*, the LlamaGen with conditional prefilling obtains 26.45 FID with the Canny edge control on ImageNet, which is much inferior to ControlNet (10.85 FID). To remedy the above issues, we formulate controllable autoregressive generation as *conditional decoding*, in which predicting the next image token is conditioned on both the previous image token and the current control token, as shown in Fig. 2 (b). Specifically, the input image token is fused with the corresponding control token and fed into the model for the next-token prediction. The proposed ControlAR leverage the conditional decoding strategy in several intermediate layers of the AR model to maintain control information across decoding layers. Fig. 2 (c) indicates that the proposed conditional decoding clearly surpasses the well-known conditional prefilling in terms of both the image quality (FID) and control capability (F1-Score). In addition, the proposed conditional decoding, without increasing the sequence length, brings negligible computation costs on the original autoregressive model, as shown in Fig. 2 (d), demonstrating superiority compared to conditional prefilling.

Most importantly, ControlAR surprisingly provides an effective way to control the resolution (size and aspect ratio) of image generation, allowing autoregressive models to get rid of the constraints of generating images at a fixed resolution, *e.g.*, LlamaGen (Sun et al., 2024) can only generate images of $256 \times 256$ after trained on $256 \times 256$ images. By adjusting the input size of the control, ControlAR decodes image tokens according to the sequence of control tokens, making it easy to achieve any-resolution image generation without resolution-aware prompts (Liu et al., 2024). In addition, we propose the multi-resolution ControlAR with multi-scale training to further enhance the image quality of different resolutions, as shown in Fig. 1.

Quantitative and qualitative experiments demonstrate that ControlAR can obtain better performance compared to previous state-of-the-art methods based on well-established diffusion models towards diverse controllable image generation. Especially, the experiments also showcase the zero-shot or fine-tuned ability to control any-resolution image generation and prove the effects of ControlAR.

The main contribution of this paper can be summarized as follows:

- We explore controllable autoregressive image generation and present ControlAR, which enables precise control and generates high-quality images. ControlAR exploits the control encoder to transform the control images into a sequence of conditional tokens and adopt the proposed conditional decoding to predict the next image token conditioned on the control and image tokens, which proves more effective than conditional prefilling.

- The proposed ControlAR easily expands autoregressive models with strong control capability. Under various control conditions, the proposed ControlAR demonstrates its highly competitive performance towards conditional consistency and image quality compared to state-of-the-art diffusion methods, *e.g.*, ControlNet++.

- We exploit the properties of our proposed conditional decoding to extend the ability of the autoregressive model to generate arbitrary resolution. With a simple multi-resolution training recipe, we extend ControlAR to Multi-Resolution ControlAR (MR-ControlAR), which allows autoregressive models to generate high-quality images with different resolutions, further enhancing their control capability.

## 2 RELATED WORK

### 2.1 IMAGE GENERATION WITH DIFFUSION MODELS

Diffusion models (Song & Ermon, 2019; Ho et al., 2020; Song et al., 2020a; Dhariwal & Nichol, 2021; Nichol et al., 2021; Lu et al., 2022; Rombach et al., 2022; Podell et al., 2023) have cemented their status as a dominant paradigm in generative modeling, especially in the domain of image synthesis. They employ an iterative denoising process to create images from Gaussian noise. Since the introduction of the diffusion model (Sohl-Dickstein et al., 2015), subsequent research has focused on refining training and sampling strategies (Song et al., 2020b; Ho et al., 2020; Song et al., 2020a). Simultaneously, in an effort to reduce computational complexity in the image generation process and enhance efficiency, numerous studies have sought to translate the generation process into the latent space (Rombach et al., 2022; Podell et al., 2023; Esser et al., 2024). Within the realm of text-to-image generation, the prevailing framework involves utilizing U-Net (Ronneberger et al., 2015)

as the denoising network, while leveraging pre-trained CLIP (Radford et al., 2021) or T5 (Raffel et al., 2020) as the text encoder to extract textual features and integrate them into the denoising process through the cross-attention mechanism. Furthermore, DiT (Peebles & Xie, 2023) employs Transformer (Vaswani, 2017) as the denoising network, yielding highly competitive results in image generation. Despite the considerable progress in diffusion models, the field of image generation still trails behind the advancement of large language models based on autoregressive mechanisms.

## 2.2 IMAGE GENERATION WITH AUTOREGRESSIVE MODELS

In contrast to the iterative denoising process of the diffusion model, the AR model operates on the principle of predicting the next image token based on the existing image tokens. Early autoregressive image generation models (Van Den Oord et al., 2016; Van den Oord et al., 2016) focused on predicting individual pixel values. Subsequent approaches (Esser et al., 2021; Ramesh et al., 2021; Yu et al., 2022) attempt to use an image tokenizer to convert continuous images into discrete tokens. More recently, there has been a growing trend towards leveraging efficient language model architectures as generative networks for AR image generation. LlamaGen (Sun et al., 2024) and Open-MAGVIT2 (Luo et al., 2024) use the Llama architecture (Touvron et al., 2023) as the generative network, demonstrating its significant potential for image generation. AiM (Li et al., 2024a) explores an approach using the Mamba model (Gu & Dao, 2023) as the generative network. Lumina-mGPT (Liu et al., 2024) develops a family of multimodal AR models capable of a wide range of visual and linguistic tasks, particularly excelling in generating flexible, photorealistic images from textual descriptions. In addition, some recent works (Xie et al., 2024; Zhou et al., 2024) fuse AR and diffusion into one multi-modal model for simultaneous image generation and understanding.

## 2.3 CONTROLLABLE IMAGE GENERATION

Relying solely on textual prompts is insufficient for conveying distinctive artistic style or precise detail during T2I image generation. Some methods (Gal et al., 2022; Ruiz et al., 2023; Zhang et al., 2023b) attempt to capture concepts from example images that are not easily described through text to guide image generation, a task known as personalization for controllable generation. Represented by ControlNet (Zhang et al., 2023a) and T2I-Adapter (Mou et al., 2024), work in this area utilizes the spatial structure of the image, such as edges, segmentation masks, depth maps, etc., to enable spatial control in the generation process. Subsequently, UniControl (Qin et al., 2023b), Uni-ControlNet (Zhao et al., 2024), and ControlNet++ (Li et al., 2024b) further extend this realm, focusing on condition encoder design and optimization of training strategies. Furthermore, Glue-Gen (Qin et al., 2023a) pairs a multi-modal encoder with a stable diffusion model for sound-to-image generation. Controllable generation based on autoregressive image generation models has been less explored. ControlVAR (Li et al., 2024c) employs next-scale prediction to jointly model control and image, but is still different from next-token prediction in autoregressive generation. Our objective is to fully harness the capabilities of autoregressive models and explore a general and efficient paradigm for controllable image generation using autoregressive models.

# 3 CONTROLAR

## 3.1 PRELIMINARY: IMAGE GENERATION WITH AUTOREGRESSIVE MODELS

Autoregressive models define the generative process as *next-token prediction*:

$$p(\mathbf{x}) = \prod_{i=1}^{n} p(x_i|x_1, x_2, \dots, x_{i-1}) = \prod_{i=1}^{n} p(x_i|x_{<i}), \tag{1}$$

and when performing image generation, $x_i$ in Eq. 1 represents the image token. The latest autoregressive image generation models such as LlamaGen (Sun et al., 2024) and AiM (Li et al., 2024a) use vector quantization to convert compressed image patches into discrete image tokens. The process of image generation is formulated as follows:

$$p(\mathbf{q}) = \prod_{t=1}^{h \cdot w} p(q_t|q_{<t}, c), \tag{2}$$

where $q_t$ is the discretised image token, $c$ is class label embedding or text embedding, and $h \cdot w$ is the total number of image tokens. During training, these two methods use causal Transformer (Vaswani, 2017) and Mamba (Gu & Dao, 2023) to model the sequence respectively, with the aim of minimising the prediction loss of the next image token, which can be written as follows:

$$\mathcal{L}_{train} = \boldsymbol{CE}(\mathbf{M}([c, q_1, q_2, \ldots, q_{l-1}]), [q_1, q_2, \ldots, q_l]), \tag{3}$$

where $\boldsymbol{CE}$ denotes cross-entropy loss, $\mathbf{M}$ denotes the sequence model, and $l$ is the sequence length.

## 3.2 UNIFIED CONDITIONAL DECODING

Autoregressive image generation models leverage the two-phase generation process, including the prefilling and decoding (Pope et al., 2023; Kwon et al., 2023), where prefilling processes the prompt tokens (or control tokens) and stores them in the KV Cache (Pope et al., 2023), and then decoding follows next-token prediction and aims to generate the output tokens (*e.g.*, image tokens). In ControlAR, we bring the condition into the decoding phase by adding the control condition token to the image token, which we refer to as conditional decoding. Specifically, we describe it as follows:

$$S_{out} = \mathcal{F}(S_{in} + \mathcal{P}(C)) = \mathcal{F}([c + C_1, I_1 + C_2, I_2 + C_3, ..., I_{i-1} + C_i]), \tag{4}$$

where $\mathcal{F}$ represents a single sequence layer modeling process in the generative network, $\mathcal{P}$ is the projection function, $S_{in}$ and $S_{out}$ are the input sequence and output sequence of each layer respectively, c is the class or text token, $I_i$ is the image token, and $C$ is the control condition sequence. It is worth noting that we use displacement by one position when adding the control condition tokens to the sequence, which allows the model to make autoregressive predictions with control information corresponding to the next image token.

Conditional decoding avoids the network having to learn the positional correspondence between the condition signal and the image, as the positional information is fixed into the sequence during the fusion of the control condition tokens. Additionally, the computational increase of this approach to the generation process is minimal. Inputting conditional signals by prefilling additional tokens will result in a significant increase in computational complexity, especially when the computational complexity of the sequence model is quadratic to the length of the sequence, as in the case of the Transformer (Vaswani, 2017). The results in Fig. 2 (d) demonstrate this.

## 3.3 CONTROLLABLE AUTOREGRESSIVE MODEL

**Overall architecture.** In our ControlAR framework shown in Fig. 3, controllable generation occurs in two main steps. First, we employ a control encoder to extract features from the control images, such as hed edges, to generate a control condition sequence of length $L$, as depicted in Fig. 3 (a). The second step involves integrating the control condition tokens into the autoregressive image generation process, as shown in Fig. 3 (b). To achieve this, we expand the sequence layer (*e.g.*, causal Transformer layer or Mamba layer) of the autoregressive model to *conditional sequence layer* by directly adding the control condition tokens to the image tokens based on positional correspondence, as discussed in Sec. 3.2. Specifically, we adopt a MLP to project the control tokens and then fuse them with image tokens via the simple addition. Furthermore, to strengthen the control of the conditions over the generated images, we evenly replace the conditional sequence layer three times in the autoregressive model. Throughout the training process of our ControlAR, we update the parameters of the sequence model, thereby enhancing the model's capability for stronger and more controllable image generation.

**Control encoder.** We propose a lightweight control encoder to transform control image to control condition tokens. In contrast to previous approaches such as ControlNet (Zhang et al., 2023a) and T2I-Adapter (Mou et al., 2024), we utilize the Vision Transformer (ViT) (Dosovitskiy, 2020) for feature extraction of control images. We believe that a ViT model, pre-trained on a large amount of data, is more adept at modeling sequences than a randomly initialized CNN network. For the class-to-image task on ImageNet, we use ViT-S to initialize our control encoder. Additionally, for the text-to-image task, we employ DINOv2-S (Oquab et al., 2023) as the initialization scheme for the control encoder. Further details on this are available in section 4.3. Notably, our ControlAR achieves efficient controllable generation with a control encoder comprising only about 22M parameters, resulting in an additional computational effort of only about 0.05T MACs for 512×512 resolution.

### 3.4 Autoregressive arbitrary-resolution generation.

Benefiting from the proposed conditional decoding, which generates the next image token conditioned on the current control token and the number of image tokens aligns with the control tokens. Therefore, we can directly adjust the resolutions of generated images according to the length of the control tokens, allowing the autoregressive models to generate arbitrary-resolution images. Rather than resizing the control images into a fixed resolution, *e.g.*, $512 \times 512$, we can directly input the control images with original resolutions into ControlAR to obtain the generated images. To further enhance the image quality of arbitrary-resolution image generation, we adopt a multi-resolution training recipe, which randomly samples different resolutions, and present the Multi-Resolution ControlAR (MR-ControlAR). Without extra modules or parameters, our MR-ControlAR is capable of generating image of arbitrary resolutions without significant quality degradation, thereby further expanding the versatility of autoregressive models.

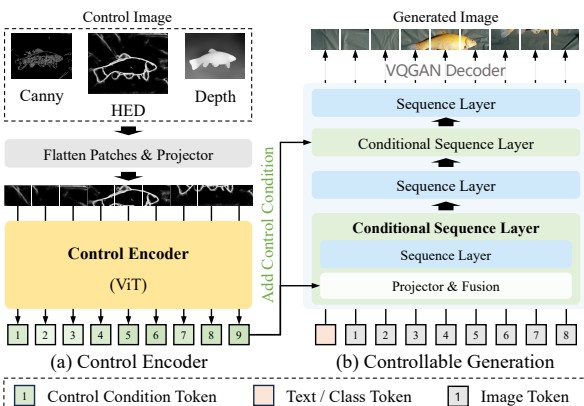

Figure 3: **The overall architecture of ControlAR**. The control image will be flattened into patches and encoded as a sequence of control tokens via the proposed **control encoder**. For controllable image generation, we extend several sequential layers (*i.e.*, causal Transformer layer or Mamba layer) of the autoregressive model into **conditional sequential layers** by incorporating the fusion of control tokens and image tokens to predict the next image token. Finally, the image tokens are decoded into a generated image through the VQGAN decoder.

## 4 Experiments

### 4.1 Experimental Setup

**Datasets.** Our experiments are divided into two main parts: class-to-image (C2I) and text-to-image (T2I) controllable generation. For the former, we follow ControlNet (Zhang et al., 2023a) to extract the canny edges and depth maps of the images in ImageNet (Deng et al., 2009) for training. In T2I experiments, we train controllable generation for segmentation masks, canny edges, hed edges, lineart edges, and depth maps. For segmentation masks, we use ADE20K (Zhou et al., 2017) and CO-COStuff (Caesar et al., 2018) as training data, with the text captions sourced from ControlNet++ (Li et al., 2024b), which adopts MiniGPT-4 (Zhu et al., 2023) to obtain a short description of the image. Furthermore, we use a subset of LAION-Aesthetics (Schuhmann et al., 2022), MultiGen-20M (Qin et al., 2023b), as the training data for canny edge, hed edge, lineart edge, and depth map controllable generation. Additional details are provided in the supplementary material.

**Evaluation and metrics.** We train the proposed ControlAR for different controllable generation tasks on several datasets and evaluate them using the corresponding validation datasets. We mainly employ two metrics: conditional consistency and Fréchet Inception Distance (FID) (Heusel et al., 2017). We evaluate the conditional consistency by calculating the similarity between the input condition images and the extracted condition images from the generated images. When evaluating segmentation masks control, we use a segmentation method, *i.e.*, Mask2Former (Cheng et al., 2022), to compute the mean Intersection-over-Union (mIoU) on generated images. We adopt the F1-Score and Root Mean Square Error (RMSE) to evaluate the similarity of canny edges and depth maps, respectively. Additionally, for hed edge and lineart edge, we utilize SSIM as the metric. Alongside these quantitative metrics, we provide abundant qualitative visualizations on diverse controls.

**Implementation details.** In C2I controllable generation experiments, we employ LlamaGen (Sun et al., 2024) and AiM (Li et al., 2024a) as the foundational autoregressive models for ControlAR. During the fine-tuning on ImageNet (Deng et al., 2009), we adopt the AdamW optimizer (Kingma,

Table 1: **C2I controllable generation.** Param. denotes the number of parameters of the C2I model. "↑" or "↓" indicate lower or higher values are better. "*" indicates that ControlVAR's FID values are estimated from its histograms (Li et al., 2024c). The results are conducted on $256 \times 256$ resolution.

| Method | C2I Model | Param. | Canny | | Depth | |
|---|---|---|---|---|---|---|
| | | | F1-Score ↑ | FID ↓ | RMSE ↓ | FID ↓ |
| ControlVAR* | VAR-d16 | 310M | - | 16.20 | - | 13.80 |
| | VAR-d20 | 600M | - | 13.00 | - | 13.40 |
| | VAR-d24 | 1.0B | - | 15.70 | - | 12.50 |
| | VAR-d30 | 2.0B | - | 7.85 | - | 6.50 |
| Ours | AiM-L | 350M | 30.36 | 9.66 | 35.01 | 7.39 |
| | LlamaGen-B | 111M | 34.15 | 10.64 | 32.41 | 6.67 |
| | LlamaGen-L | 343M | 34.91 | 7.69 | 31.11 | 4.19 |

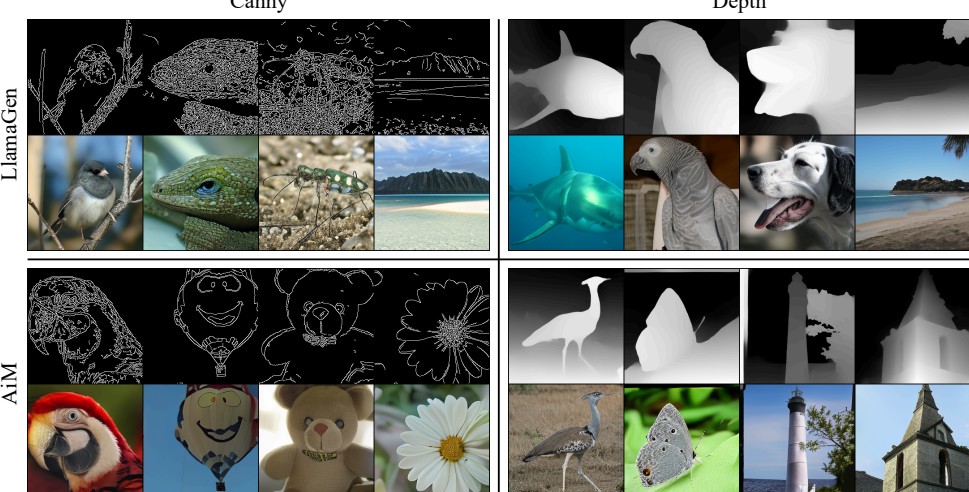

Figure 4: **Visualization of C2I controllable generation.** Our ControlAR generates images with high conditional consistency and quality on both LlamaGen and AiM.

2014). The learning rate is set to 1e-4 and 8e-4 for training LlamaGen and AiM respectively. We use the image size of $256 \times 256$, with a batch size of 256 for canny edge and depth maps. In T2I experiments, we mainly use LlamaGen-XL, which is built upon a T5 encoder (Raffel et al., 2020) and contains 775M parameters. We employ the AdamW optimizer with a learning rate of 5e-5 and resize both input and control images to $512 \times 512$ for comparison with other methods.

## 4.2 EXPERIMENTAL RESULTS

**C2I controllable generation.** We utilize the ImageNet (Deng et al., 2009) to conduct controllable generation experiments for C2I, and the results are shown in Tab. 1. We calculate the conditional consistency (F1-Score or RMSE) and the FID of the images generated by ControlAR and compare the FID with ControlVAR (Li et al., 2024c). It shows that our proposed ControlAR achieves lower FID based on the LlamaGen-L, which only has 16.7% of parameters of VAR-d30 (Tian et al., 2024). In addition, the experimental results show that our method achieves good results with different autoregressive models including Transformer-based LlamaGen and Mamba-based AiM. Fig. 4 illustrates the visualizations of ControlAR with different autoregressive models.

**T2I controllable generation.** We mainly employ LlamaGen-XL as the autoregressive model for T2I generation. Tab. 2 presents the quantitative comparison of controllability with state-of-the-art methods. Among those methods in Tab. 2, GLIGEN (Li et al., 2023) utilizes SD1.4 (Rombach et al., 2022) as the generative model, while T2I-Adapter (Mou et al., 2024), Uni-ControlNet (Zhao et al., 2024), UniControl (Qin et al., 2023b), ControlNet (Zhang et al., 2023a), and ControlNet++ (Li et al., 2024b) adopt SD1.5 (Rombach et al., 2022) as the generative model. As Tab. 2 shows, it is evident that our ControlAR is highly competitive compared to the existing diffusion-based methods. The proposed ControlAR significantly outperforms ControlNet (Zhang et al., 2023a) in terms

Table 2: **Conditional consistency of T2I controllable generation.** "↑" or "↓" indicate lower or higher values are better. "-" denotes that the method does not release a model for testing. The results are conducted on $512 \times 512$ resolution.

| Method | Seg. | | Canny | Hed | Lineart | Depth |
|---|---|---|---|---|---|---|
| | mIoU ↑ | mIoU ↑ | F1-Score ↑ | SSIM ↑ | SSIM ↑ | RMSE ↓ |
| | ADE20K | COCOStuff | MultiGen-20M | MultiGen-20M | MultiGen-20M | MultiGen-20M |
| GLIGEN | 23.78 | - | 26.94 | - | - | 38.83 |
| T2I-Adapter | 12.61 | - | 23.65 | - | - | 48.40 |
| Uni-ControlNet | 19.39 | - | 27.32 | 69.10 | - | 40.65 |
| UniControl | 25.44 | - | 30.82 | 79.69 | - | 39.18 |
| ControlNet | 32.55 | 27.46 | 34.65 | 76.21 | 70.54 | 35.90 |
| ControlNet++ | **43.64** | 34.56 | 37.04 | 80.97 | **83.99** | **28.32** |
| Ours | 39.95 | **37.49** | **37.08** | **85.63** | 79.22 | 29.01 |

Table 3: **FID of T2I controllable generation.** "-" denotes that the method does not release a model for testing. Our ControlAR achieves significant FID improvements.

| Method | Seg. | | Canny | Hed | Lineart | Depth |
|---|---|---|---|---|---|---|
| | ADE20K | COCOStuff | MultiGen-20M | MultiGen-20M | MultiGen-20M | MultiGen-20M |
| GLIGEN | 33.02 | - | 18.89 | - | - | 18.36 |
| T2I-Adapter | 39.15 | - | 15.96 | - | - | 22.52 |
| Uni-ControlNet | 39.70 | - | 17.14 | 17.08 | - | 20.27 |
| UniControl | 46.34 | - | 19.94 | 15.99 | - | 18.66 |
| ControlNet | 33.28 | 21.33 | **14.73** | 15.41 | 17.44 | 17.76 |
| ControlNet++ | 29.49 | 19.29 | 18.23 | 15.01 | 13.88 | 16.66 |
| Ours | **27.15** | **14.51** | 17.51 | **10.53** | **12.41** | **14.61** |

of diverse control tasks. Compared to ControlNet++ (Li et al., 2024b), which is fine-tuned based on the well-established ControlNet, our ControlAR demonstrates comparable or even better controlling performance, for example, achieving an improvement of 4.66 SSIM on the hed edges task. Additionally, we report the FID for the generated images in Tab. 3. Our approach attains the better FID across various tasks compared with ControlNet++, indicating that it not only possesses strong controllability but also ensures the quality of image generation. We provide qualitative comparison in Fig. 5, and more visualizations are available in the supplementary material.

**Arbitrary-Resolution Generation.** Instead of uniformly resizing the controls and images to 512 × 512, we adopt a set of resolutions to train our Multi-Resolution ControlAR. Specifically, we randomly sample the height and width of the training data from 384 to 1024 with a minimum interval of 16, and the image can be resized when it satisfies (H/16)×(W/16)≤2304. In addition, we need to adjust the parameter settings of the rotational position encoding in the generative network by simply increasing its maximum sequence length to 2304. Direct end-to-end controllable generation using MR-ControlAR preserves the detailed features of the control image and avoids the loss of information due to scaling. We show the difference between these two approaches in Fig. 6 (a), and perform hed edge control generation experiments on images with different resolution ratios in the validation set of MultiGen-20M. Experimental results show that after multi-resolution training, MR-ControlAR can ensure that the generation of images with different resolution ratios is not impaired. We show the visualization at different resolutions in Fig. 1.

## 4.3 ABLATION STUDIES

**Ablations on the Control Encoder.** In Tab. 4, we conduct experiments using different encoders (or pre-training schemes) towards different controls, including canny edge, depth map, and hed edge. Firstly, we follow T2I-Adapter (Mou et al., 2024) and design a vanilla convolutional control encoder with 4 consecutive residual blocks (He et al., 2016) and a total downsample ratio of 16. The vanilla CNN-based control encoder contains 21.8M parameters, which has similar parameters with ViT-S (Dosovitskiy, 2020). Further, we explore the effects of using pre-trained ViTs as our control encoder and adopt ViTs with different pre-trained schemes, *i.e.*, the ImageNet-supervised (Dosovitskiy, 2020) and self-supervised (Oquab et al., 2023). For our experiments, we employ LlamaGen-B's C2I model to conduct experiments on the ImageNet (Deng et al., 2009) for canny edge and depth

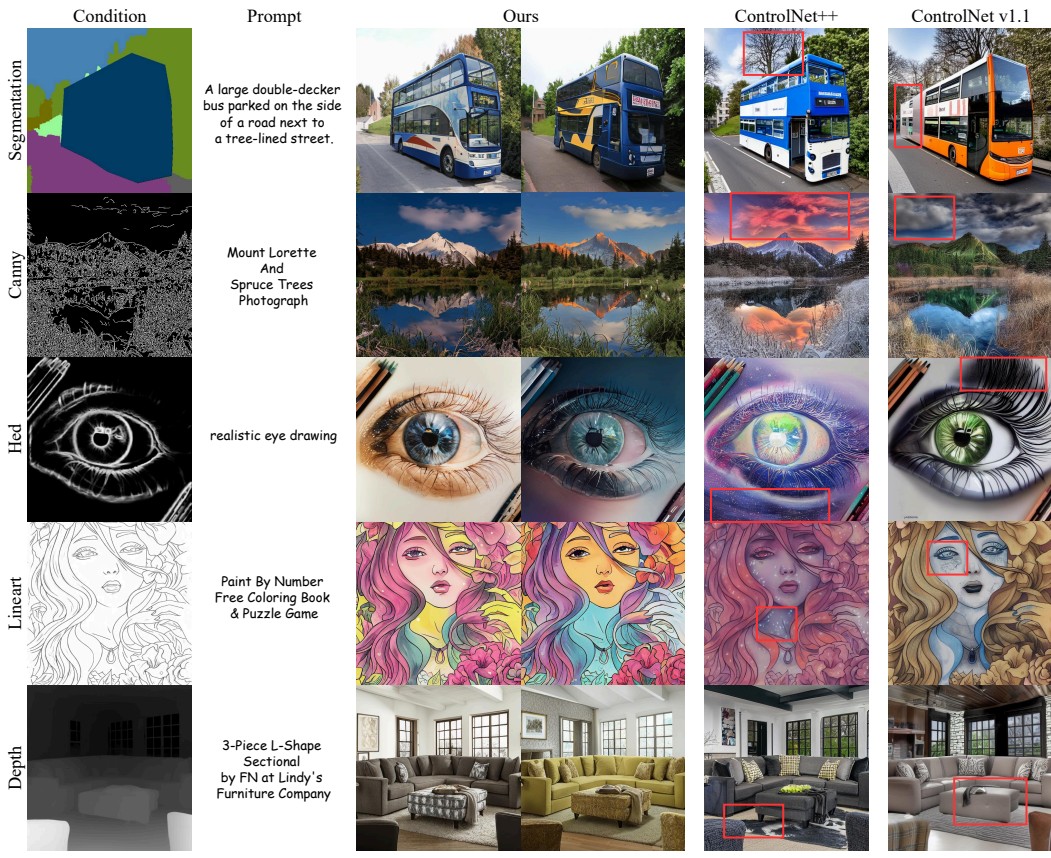

Figure 5: **Visualization of text-to-image controllable generation.** We use red boxes to mark areas where the generated results of other methods differ from the input control image.

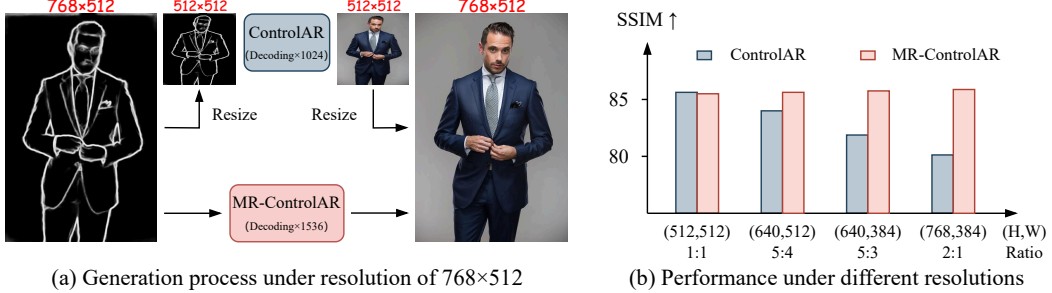

(a) Generation process under resolution of 768×512        (b) Performance under different resolutions

Figure 6: **Comparison of ControlAR and Multi-Resolution ControlAR.** (a) shows the generation process of ControlAR and MR-ControlAR under the resolution of $768 \times 512$. "Decoding $\times$ 1024" denotes that 1024 image tokens need to be decoded for output. (b) compares the conditional consistency of ControlAR and MR-ControlAR under different resolutions of control conditions.

map conditions, while using LlamaGen-XL's T2I model to carry out experiments on the MultiGen-20M (Qin et al., 2023b) for the hed edge condition. As depicted in the table, different control encoders exhibit varying performance across different datasets. The reason for this phenomenon is the different pre-training data for the two models. ViT-s is obtained by pre-training on ImageNet and thus is more advantageous for C2I tasks that are also trained on ImageNet. DINOv2-s, on the other hand, is pre-trained on a larger and more diverse data such as LVD-142M, and thus will be more suitable for T2I tasks trained on MultiGen20M, which is also a diverse text-image paired dataset. Furthermore, we evaluate the performance of larger encoders such as DINOv2-B, and the outcomes reveal that higher parameter counts enable our method to achieve superior results.

Table 4: **Ablations on the Control Encoder.** "↑" or "↓" indicate lower or higher values are better.

| Control Encoder | Params | Canny (C2I) | | Depth (C2I) | | Hed (T2I) | |
|---|---|---|---|---|---|---|---|
| | | F1-Score ↑ | FID ↓ | RMSE ↓ | FID ↓ | SSIM ↑ | FID ↓ |
| CNN (4× Res. Blocks) | 21.8M | 33.55 | 12.27 | 33.36 | 6.97 | 81.64 | 15.33 |
| ViT-S | 22.1M | **34.15** | 10.64 | 32.41 | 6.64 | 82.37 | 14.59 |
| DINOv2-S | 22.1M | 33.38 | 10.87 | 32.82 | 7.31 | 85.63 | 10.53 |
| DINOv2-B | 86.6M | 34.07 | **9.47** | **31.81** | **6.36** | **86.12** | **8.58** |

Table 5: **Ablations on the control fusion strategy.** "↑" or "↓" indicate lower or higher values are better.

| Fusion Strategy | #Layer | F1-Score ↑ | FID ↓ |
|---|---|---|---|
| Cross-Attention | 1-th | 30.86 | 15.34 |
| Addition | 1-th | 34.01 | 11.02 |
| Addition | 1,5,9-th | 34.15 | 10.64 |
| Addition | 1∼12-th | 34.21 | 11.75 |

Table 6: **Ablations on the training strategy.** "↑" or "↓" indicate lower or higher values are better.

| Training Strategy | F1-Score ↑ | FID ↓ |
|---|---|---|
| Freeze | 30.62 | 13.67 |
| LoRA | 32.90 | 13.20 |
| Full fine-tune | 34.15 | 10.64 |

**Ablations on the Control Fusion Strategy.** We explore different strategies for fusing control condition tokens with image tokens using the canny edge condition on ImageNet and LlamaGen-B as the generative model. Tab. 5 shows the results. Specifically, when using cross-attention for control fusion, we assign control condition tokens as the key and value, while image tokens serve as the query. Within LlamaGen-B, consisting of 12 layers of Transformer, we conduct experiments with addition at the first layer, addition at layer 1, 5, and 9, and addition at each layer. The results indicate that direct addition proves more efficacious than cross-attention. This outcome may be due to cross-attention needing to first understand the positional relationship between the image block and the control condition token, potentially leading to slower convergence. Furthermore, augmenting the frequency of addition yields enhanced conditional coherence within the generated imagery. However, an excessive degree of addition also correlates with an increase in FID.

**Ablations on Sequence Model Training Strategy.** We conduct ablation experiments on the parameter update strategy of the sequence model during training. In the field of controllable generation, the most common ways of updating the parameters of a generative model include complete freezing, updating using Low-Rank Adaptation (LoRA) (Hu et al., 2021), and full fine-tuning. The results of the experiment are displayed in Tab. 6. We use LlamaGen-B as the generative model for experiments on ImageNet based on canny edge. Experimental results show that full fine-tuning outperforms other schemes in terms of conditional consistency and FID of the generated images.

**Conditional Decoding *v.s.* Conditional Prefilling.** In this part, we present a comprehensive comparison between two methods: conditional decoding and conditional prefilling. We use LlamaGen-B as the generative model for experiments on ImageNet based on canny edge condition. In Fig. 2 (c), we depict the conditional consistency (F1-Score) and FID with respect to the number of training epochs for both approaches. Conditional decoding exhibits significant superiority over conditional prefilling in terms of both the speed of convergence and the final convergence result. Additionally, we provide a comparison of training resource consumption between the two approaches in Fig. 2 (d). Due to the substantial increase in the length of the sequence, conditional prefilling results in heightened memory consumption during training, as well as a notable decrease in training speed.

## 5 CONCLUSION

In this paper, we address autoregressive controllable image generation and present ControlAR, which allows autoregressive models to generate high-quality images according to diverse spatial controls. The proposed ControlAR encodes the spatial controls and adopts conditional decoding to superimpose control condition tokens on the image generation process. Moreover, ControlAR extends the capability of the autoregressive image generation model for arbitrary-resolution image generation. Experimental results under a variety of control conditions show that ControlAR is capable of precise control without compromising image quality, and is also very competitive with the diffusion model-based state-of-the-art methods.

ACKNOWLEDGMENT

This work was supported by the National Natural Science Foundation of China (No. 62376102). We sincerely thank Jingfeng Yao, Lianghui Zhu, and Zhuoyan Luo for their kind and helpful discussions about the draft.

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

## A APPENDIX

### A.1 IMPLEMENTATION DETAILS

**Dataset details.** The quantity of images from all datasets utilized in our experiment is detailed in Tab. 7. We utilize the ImageNet-1K (Deng et al., 2009) as the training dataset for class-to-image controllable generation, encompassing a total of 1,000 classes. The canny edge detector (Canny, 1986) is employed to acquire the canny edge map, and the depth map is obtained using Midas (Ranftl et al., 2020). In the context of text-to-image controllable generation, ADE20K (Zhou et al., 2017) and COCOStuff (Caesar et al., 2018) are harnessed for training the segmentation control task, while MultiGen-20M is utilized for training the edge map and depth control generation.

Table 7: **Details of different dataset.**

|  | ImageNet-1K | ADE20K | COCOStuff | MultiGen-20M |
|---|---|---|---|---|
| Training Samples | 1281188 | 20210 | 118287 | 2810616 |
| Evaluation Samples | 50000 | 2000 | 5000 | 5000 |

**Evaluation details.** To assess the conditional consistency of the generated images, we have devised various metrics tailored to each specific task. In the context of segmentation control generation, we employ a segmentation model to evaluate the mean Intersection over Union (mIoU) of the generated images. Specifically, we reference ControlNet++ to examine the results of the validation set generation on ADE20K using Mask2Former (Cheng et al., 2022), and on COCOStuff using DeepLabv3 (Chen, 2017). For canny edge control generation, we utilize the canny edge detector with thresholds of (100, 200) to derive the canny edge of the results, and subsequently calculate the F1-Score in relation to the input control. In the case of hed and lineart edge, we follow the approach outlined in ControlNet to obtain control images and compute the Structural Similarity Index (SSIM). Regarding depth map control generation, we calculate the Root Mean Square Error (RMSE).

Table 8: **Training details of different tasks.**

|  | Seg. | | Canny | Hed | Lineart | Depth |
|---|---|---|---|---|---|---|
|  | ADE20K | COCOStuff | | MultiGen-20M | | |
| Batch size | 96 | 96 | 96 | 88 | 88 | 96 |
| GPU hours | 55 | 80 | 340 | 160 | 110 | 370 |

**Training details.** We use 8 Nvidia A100 80G GPUs to complete text-to-image controllable generation experiments based on LlamaGen-XL (Sun et al., 2024). The batch size settings and GPU hours during training can be found in Tab. 8. We use the edge extraction model to obtain the hed edge and lineart edge of the image during the training process, which takes up some memory, so the batch size is slightly smaller than the other tasks. It should be noted that since the ADE20K dataset has less training data, we first merge the ADE20K and COCOStuff datasets together to train the model, which requires roughly 50 GPU hours. Because the segmentation map labelling is inconsistent between the two datasets, we fine-tuned 2k iterations on ADE20K and 20k iterations on COCOStuff, respectively. The additional 2k iteration on ADE20K results in a mIoU improvement of 1.15.

### A.2 MORE EXPERIMENTAL EXPLORATIONS

**Comparison with recent work.** We have added some quantitative comparative results with recent work including OmniGen (Xiao et al., 2024) and Lumina-mGPT (Liu et al., 2024), as shown in the Tab. 9. The results for segmentation task are measured on the validation set of ADE20K (Zhou et al., 2017), and the results for canny, hed and depth are measured on the validation set of MultiGen-20M (Qin et al., 2023b). OmniGen uses iterative denoising diffusion for image generation, while lumina-mGPT uses autoregressive prediction. Although Lumina-mGPT has a much larger number of parameters than our ControlAR, it does not perform particularly well on the controllable generation task. Our ControlAR provides a good solution for autoregressive controllable image generation

and our method does not require any adjustments to the structure of the generative network or modifications to the length of the sequences, which means that we can easily migrate our ControlAR to other autoregressive image generation models, such as Lumina-mGPT.

Table 9: **Quantitative comparison with recent works.**

| Method | Param. | Seg.(mIoU↑) | Canny(F1-Score↑) | Hed(SSIM↑) | Depth(RMSE↓) |
|---|---|---|---|---|---|
| OmniGen | 3.8B | 44.23 | 35.54 | 82.37 | 28.54 |
| Lumina-mGPT | 7B | 25.02 | 29.99 | 78.21 | 55.25 |
| Ours | 0.8B | 39.95 | 37.08 | 85.63 | 29.01 |

**Adjustable control strength.** Given the diversity of image structures, we sometimes do not want the spatial structure of the generated image to be identical to the input control. To achieve this, it is only necessary to skip the operation of fusing the control condition token with the image token with a probability of 50% when training ControlAR. Such an approach ensures ControlAR's generative capability in the absence of control image inputs. At the same time, multiplying the control condition token by a control strength factor $\alpha$ during inference changes the degree of control of the generated result. When $\alpha$ is 1, ControlAR will generate an image exclusively based on the control condition, while when $\alpha$ is 0, the generated results will be related only to the text prompt. Fig. 7 shows the visualizations using edges as the control image and adjusting the control strength.

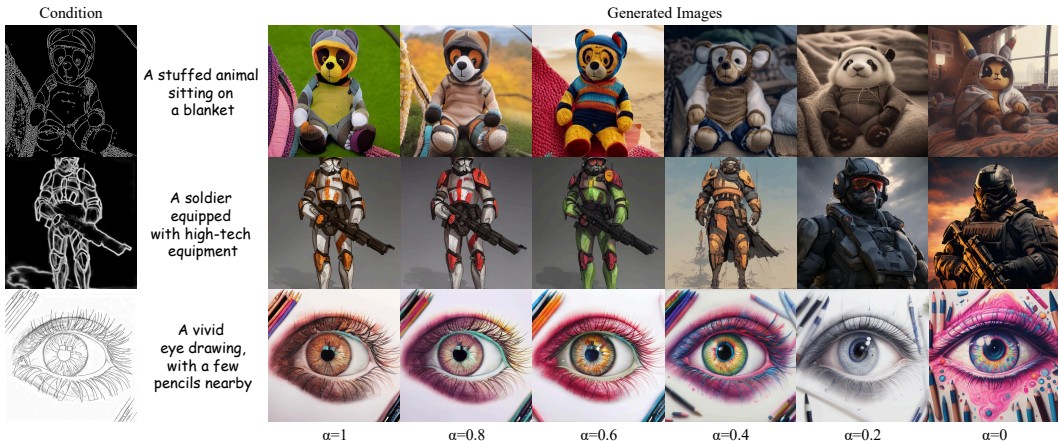

Figure 7: **Visualization with different control strength factor $\alpha$.**

**Arbitrary-Resolution Generation Without Condition Image.** We conduct a more in-depth exploratory study on resolution control in the absence of specific condition image. We can generate a grayscale map of the corresponding resolution according to the desired height and width, this grayscale map consists of a number of $16 \times 16$ small squares, and the grayscale value of each row decreases from left to right, the left most 255, the right most 0. This grayscale image is the condition image that determines the resolution. Thanks to the strong positional dependence of the control decoding strategy between the image token and the control condition token, the model only needs to generate a sequence as long as the control condition sequence. And since the grayscale value of each row is decreasing from left to right, the model can easily know when it is necessary to switch to the next row. We have verified the feasibility of this approach on a small experimental scale. We show some visualization results in Fig. 8. Using resolution-aware prompts to control the resolution as in Lumina-mGPT requires the constant generation of $< end - of - line >$ tokens during the prediction of the image and the eventual prediction of $< end - of - image >$ token. This approach requires the model to make its own decisions about where to make line breaks and where to end generation, but our ControlAR is directly telling the model where to make line breaks and end generation. We only need to fine-tune the weights based on LlamaGen-XL (512×512) on about 1M text-image paired data for 30k steps to achieve a good arbitrary resolution generation capability

without specific control image. This proves that our ControlAR can be a very effective strategy for controlling resolution.

## A.3 DISCUSSION

**Limitation.** We have shown in our experiments that updating the parameters of the generative model can achieve better results than freezing it completely. However, this approach is still not as convenient as ControlNet in terms of model portability. In addition, our method does not currently support scenarios where multiple control images are input simultaneously. Processing multiple control images simultaneously using a control encoder with a small number of parameters can be challenging.

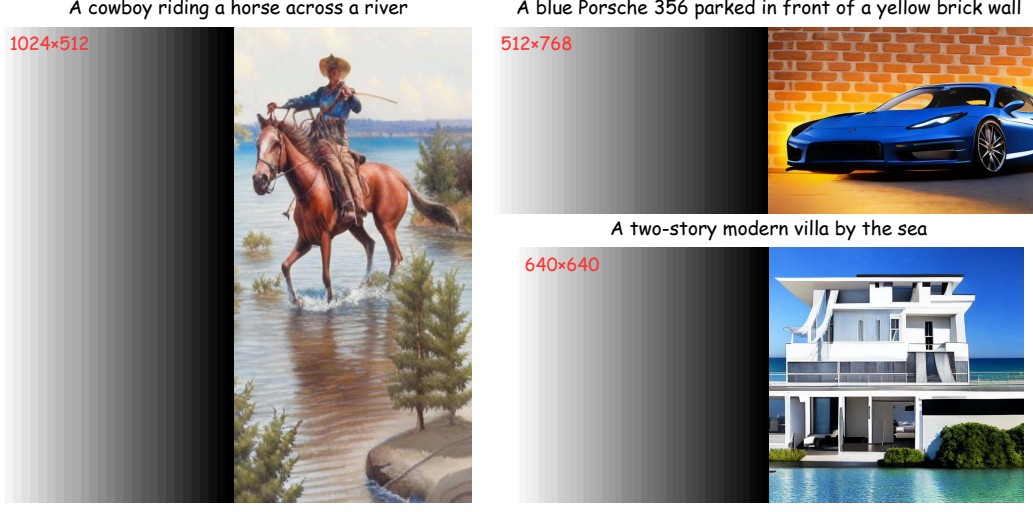

Figure 8: **Arbitrary-Resolution generation without condition image.** The grayscale map on the left is the condition image generated according to the desired resolution.

**Failure Cases.** ControlAR performs well on the conditional consistency of controllable generation of spatial structures. But because of this, the generated images are sometimes less controlled by the text prompt, especially when the textual prompt conflicts with the spatial structure of the control image. We use depth-to-image and canny-to-image as examples in Fig. 9. When there is a large difference between the text prompt and the original image, it might fail to generate images according to the text prompt. In ControlAR, we can use the control factor to adjust the strength of spatial control, thereby aligning the generated results with the text and mitigating this conflict. However, the conflict between text prompts and spatial controls is a common issue in current control-to-image generation models, including ControlNet (Zhang et al., 2023a) and ControlNet++ (Li et al., 2024b). As shown in Fig. 9, neither ControlNet nor ControlNet++ can generate images that strictly follow the text prompts. Moreover, ControlNet++ introduces additional supervision to facilitate alignment between the generated image and spatial controls, which weakens the influence of the text prompt as shown in the case of canny-to-image. This phenomenon reflects that there may be some confrontation between the structural freedom of the generated image and the conditional consistency.

The examples in Fig. 9 reflect the possible contradiction between structure diversity and conditional consistency. We acknowledge that structure diversity is a meaningful and challenging problem for controllable image generation. We extend ControlAR by introducing a control $\alpha$ to dynamically adjust the strength of control. This allows the model to balance structural consistency and diversity, enabling the generated images to align with the input geometric controls while also introducing variations to produce richer and more diverse structures. Although this is an exploratory attempt, we believe that ControlAR has the potential to achieve this balance. Specifically speaking, in order to improve the diversity of generated images, we believe that we need to explore suitable training strategies to achieve the effect of being able to adjust the intensity of control during the inference

phase, and to resolve the possible contradiction between text alignment and conditional consistency, which are important directions in future research.

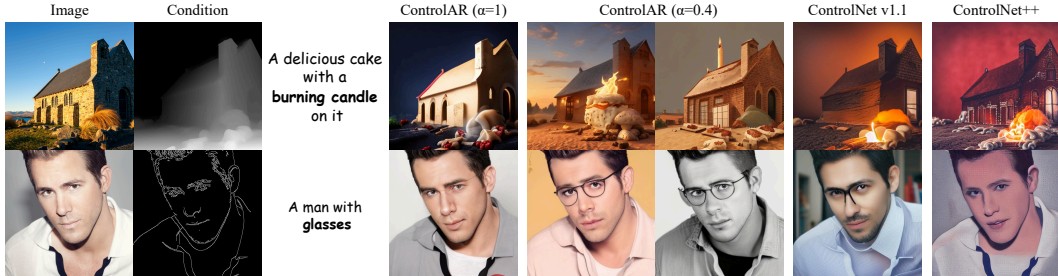

Figure 9: **Failure cases of current ControlAR.** When the text prompt conflicts with the control image, the generated result tends to ignore the text prompt. Adjusting the control strength factor $\alpha$ can alleviate this problem.

**Future work.** We will use more data to try more kinds of conditional control generation, such as human pose and bounding box. At the same time, in order to improve the migratability of the model we will consider focusing the parameter update on the control encoder and keep the parameters of the generated model itself unchanged. In addition to this, how to use one control encoder to process different control image inputs simultaneously is also a direction worth exploring.

## A.4 MORE VISUALIZATIONS

More visualization results under different conditions of control are shown in Fig. 10 11 12 13 14. We alse show some visualization comparison of ControlAR and MR-ControlAR at different resolution in Fig. 15 and Fig. 16.

Condition             Generated Images

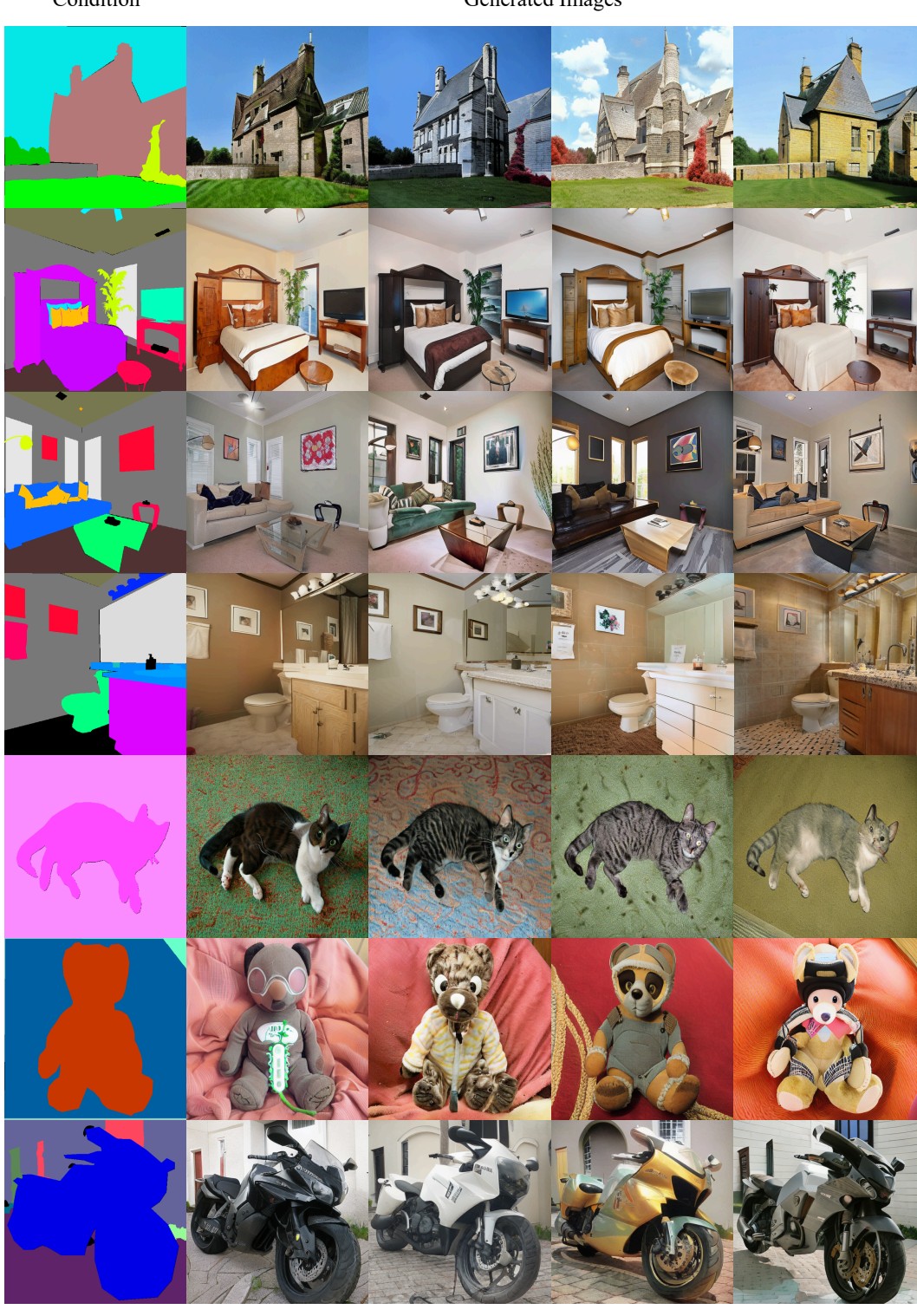

Figure 10: **Segmentation mask control generation visualization.**

Condition                                  Generated Images

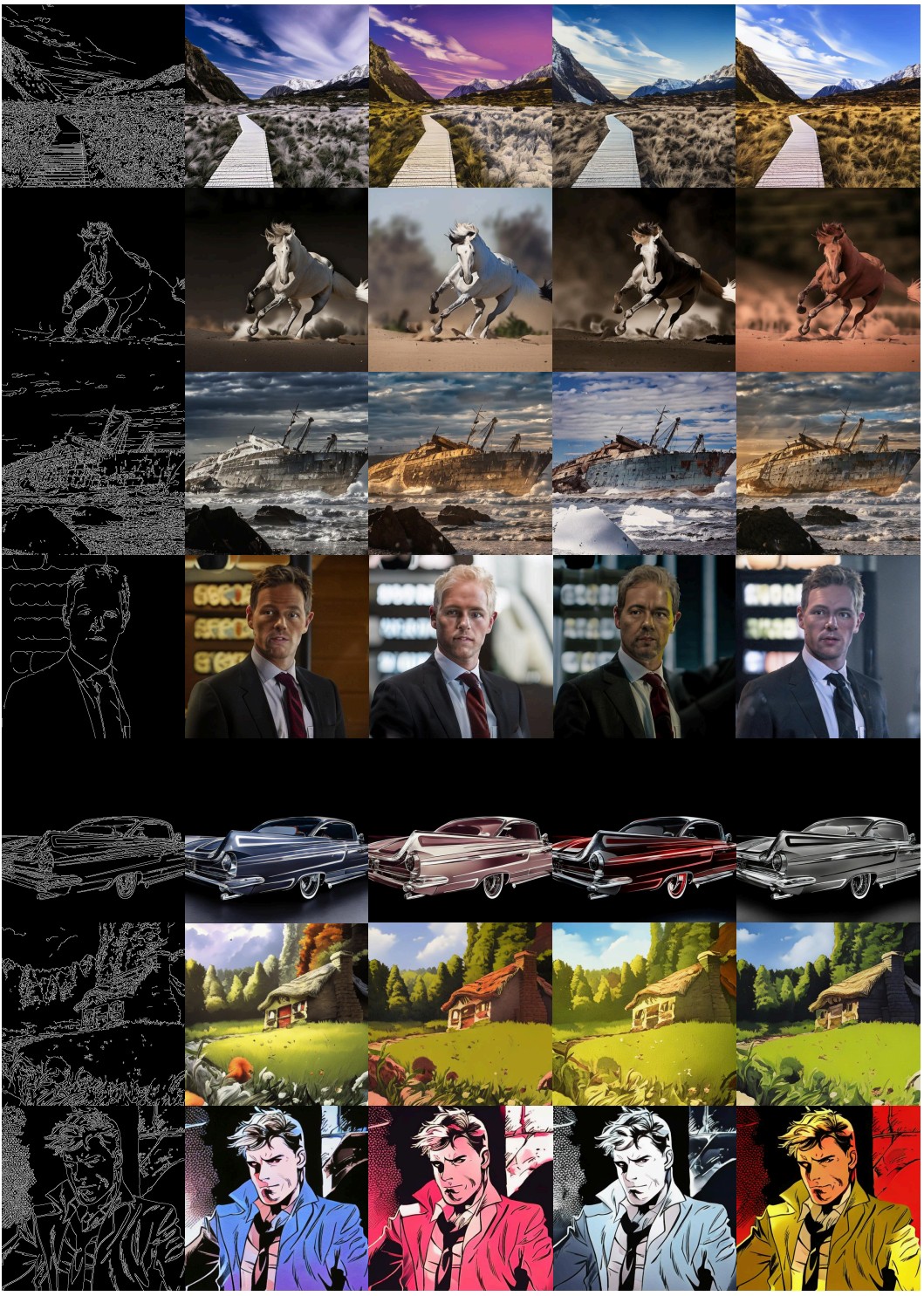

Figure 11: **Canny edge control generation visualization.**

Condition                          Generated Images

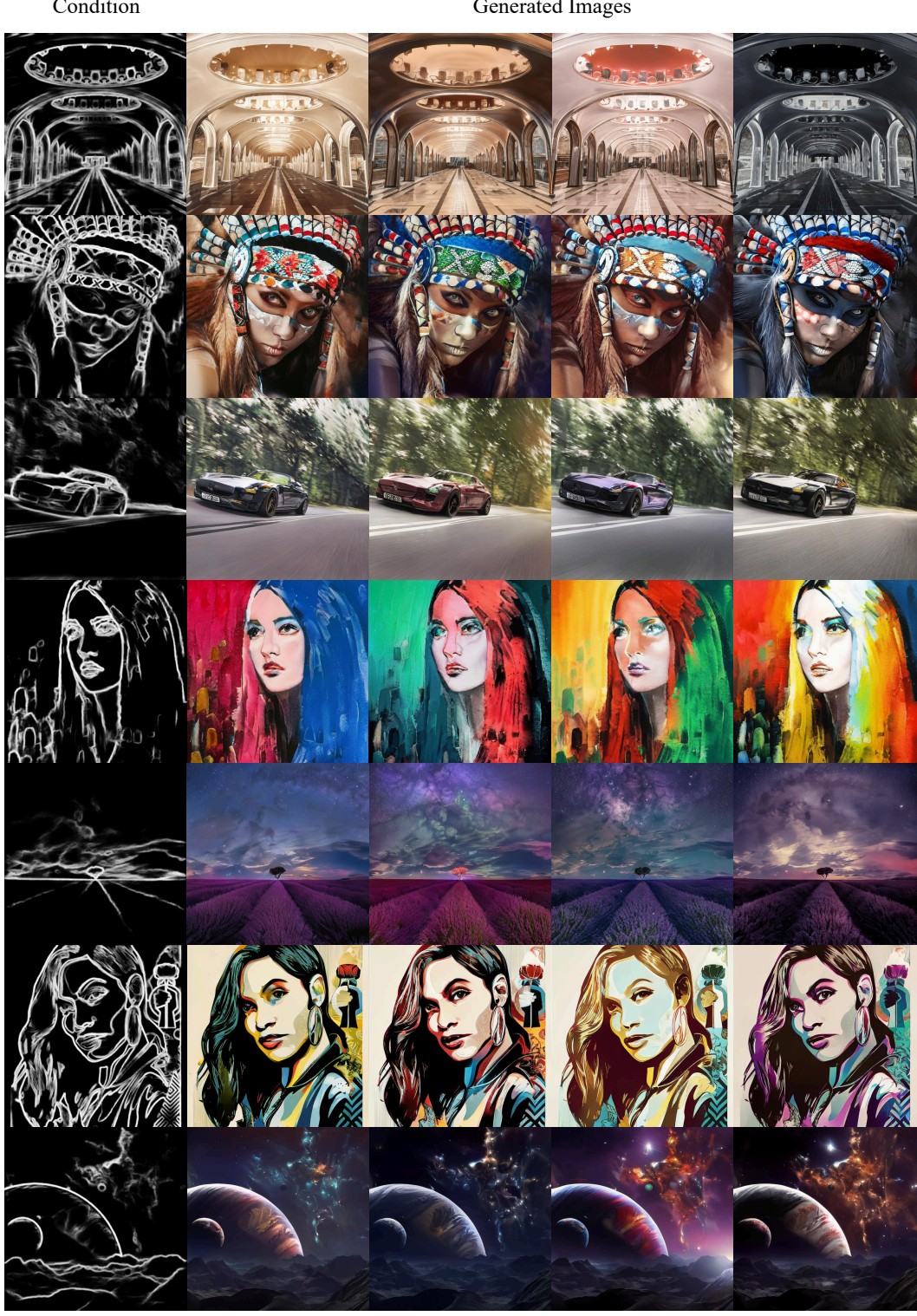

Figure 12: **Hed edge control generation visualization.**

Condition                                    Generated Images

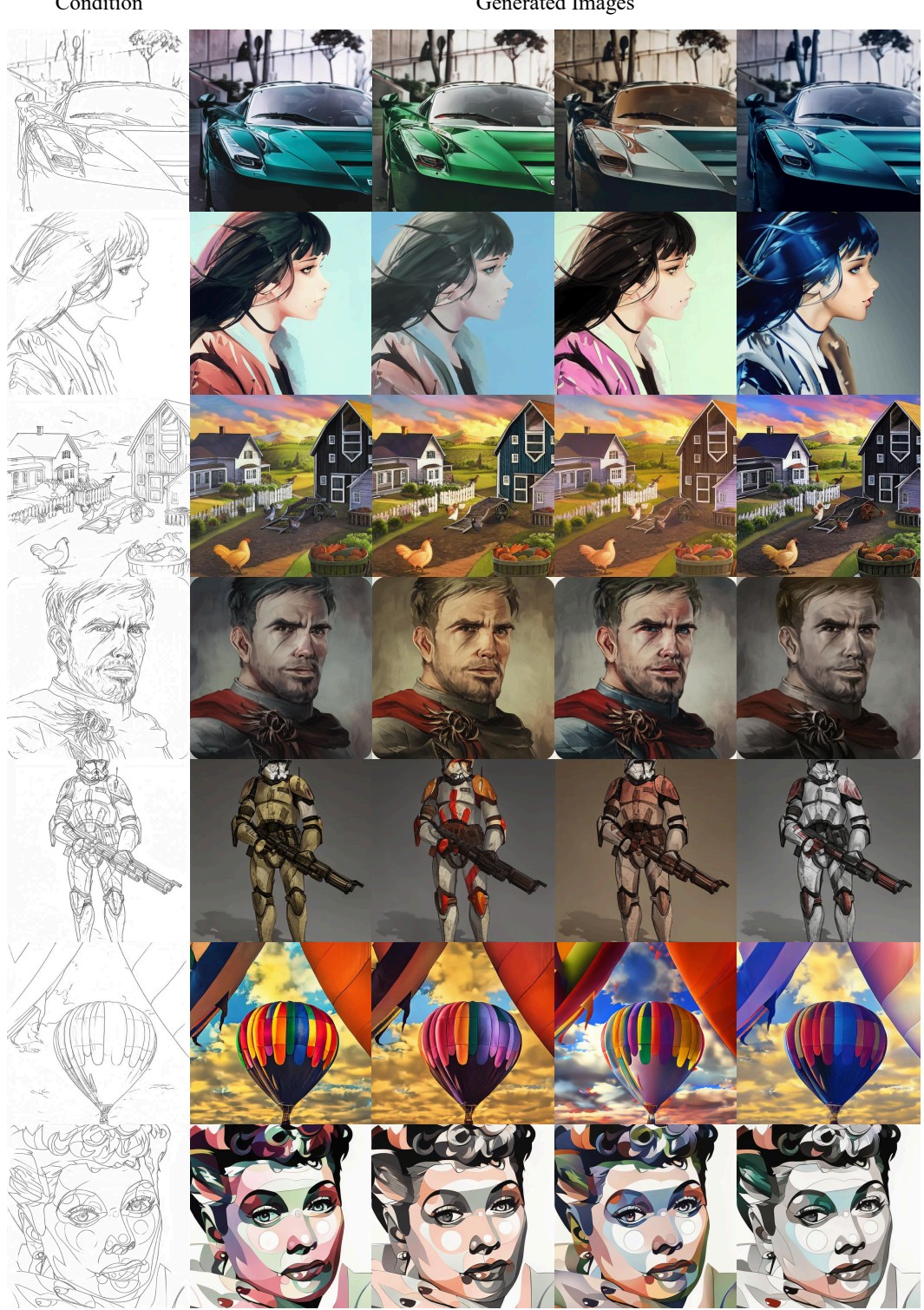

Figure 13: **Lineart edge control generation visualization.**

Condition                                    Generated Images

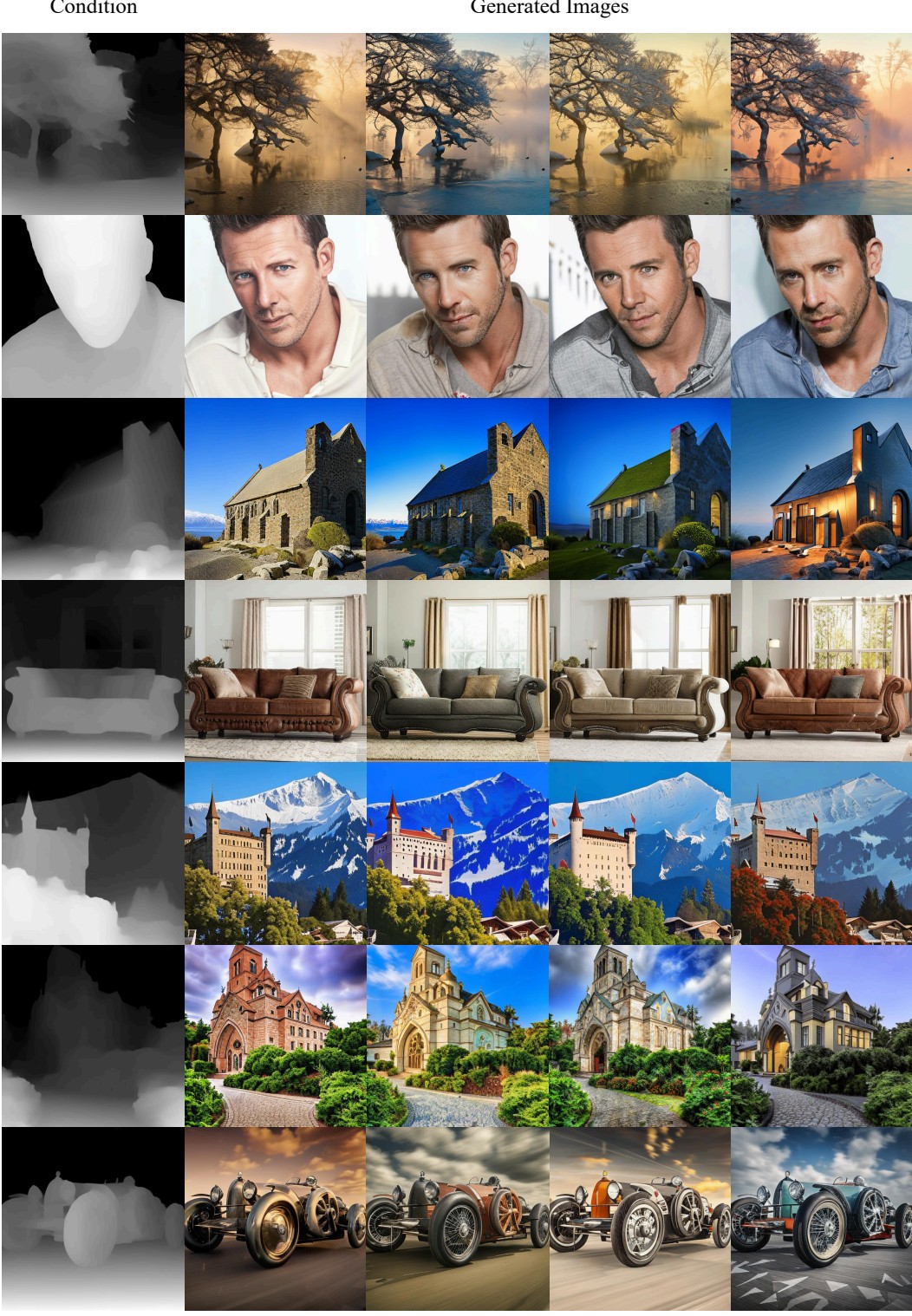

Figure 14: **Depth map control generation visualization.**

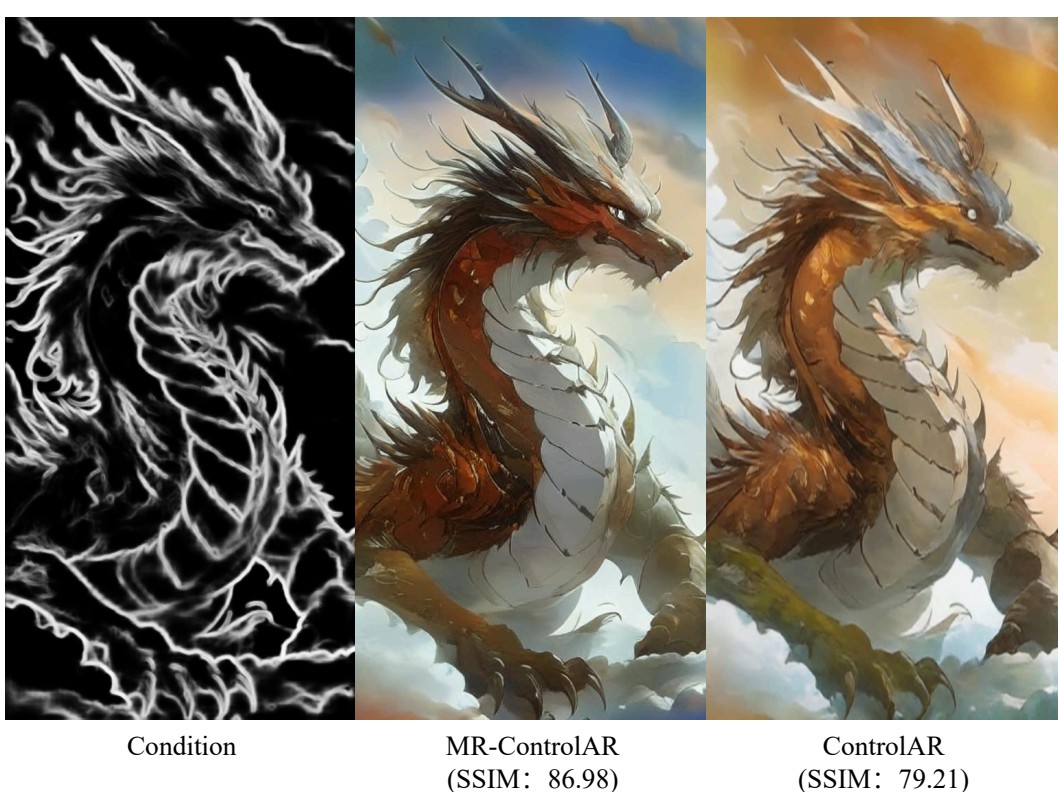

| Condition | MR-ControlAR | ControlAR |
|:---:|:---:|:---:|
| | (SSIM：86.98) | (SSIM：79.21) |

Figure 15: **visualization comparison of MR-ControlAR and ControlAR at the resolution of** $1024 \times 512$**.**

Condition

MR-ControlAR
(SSIM：83.38)

ControlAR
(SSIM：78.82)

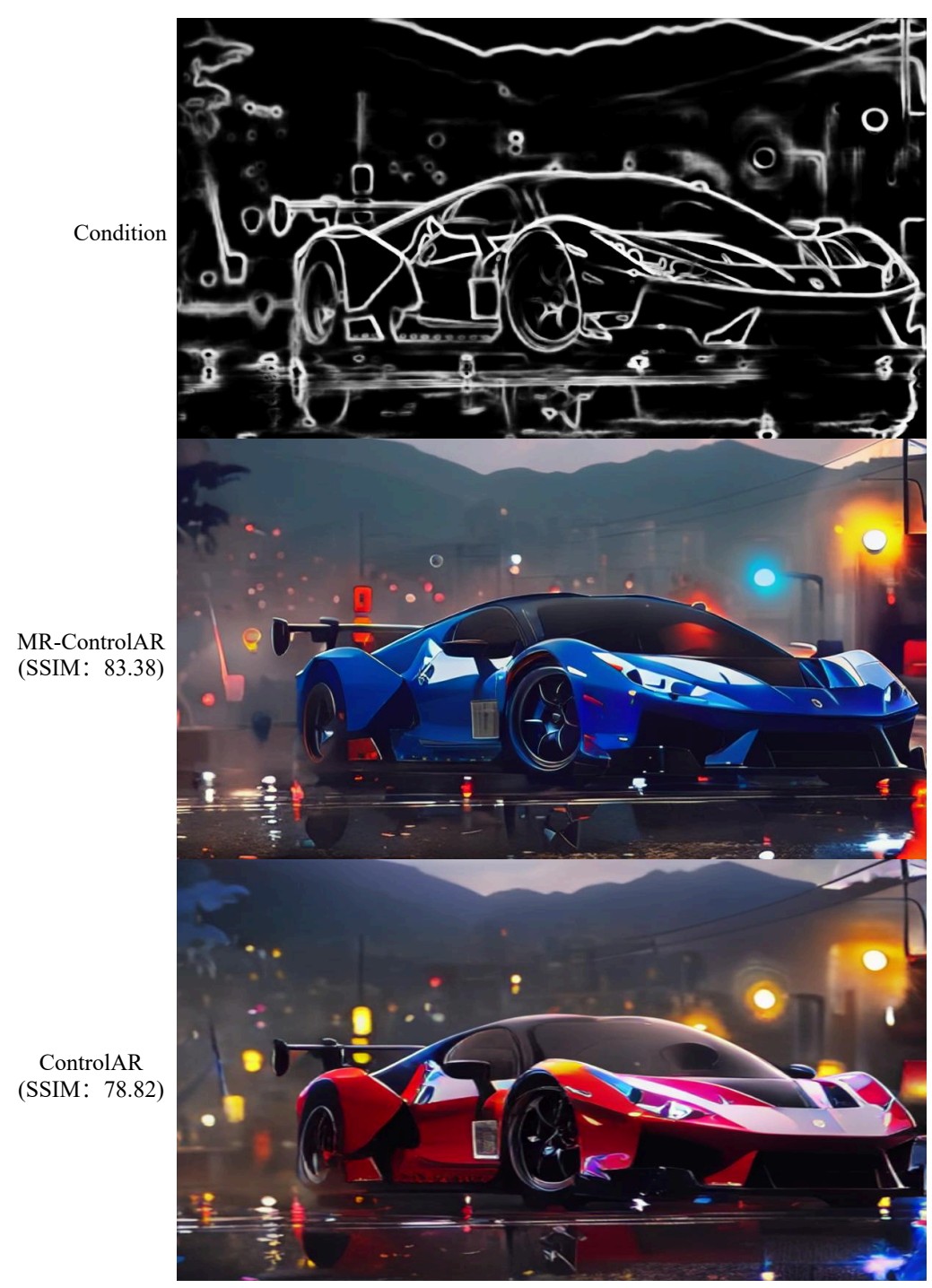

Figure 16: **visualization comparison of MR-ControlAR and ControlAR at the resolution of** $576 \times 1024$**.**

