# OpenReview forum: "ControlAR: Controllable Image Generation with Autoregressive Models"
_ICLR.cc/2025/Conference — ICLR 2025 Poster_

### Official Review · Reviewer_Scc6 · 2024-11-02

**Soundness:** 3
**Presentation:** 3
**Contribution:** 3
**Rating:** 5
**Confidence:** 4

**Summary:**

This paper introduces ControlAR, a framework that enables autoregressive (AR) models to generate high-quality images with precise spatial controls.

**Strengths:**

1. The idea is simple but effective: a lightweight control encoder that transforms spatial inputs (edges, depth maps, etc.) into control tokens, and conditional decoding method that fuses control tokens with image tokens during generation.

2. This work shows strong results across multiple tasks (edges, depth maps, segmentation).

**Weaknesses:**

1. The paper's efficiency comparison between ControlAR and ControlNet++ (22M vs 361M parameters) is misleading. Comparing parameter counts between diffusion-based and AR-based models is fundamentally unfair due to their different architectures and generation mechanisms. The paper should instead compare inference time, FLOPs, or other more relevant efficiency metrics between similar architectures.


2. The use of pretrained visual encoders (CLIP, DINOv2) is a standard practice in multimodal learning

3. The quantitative improvements shown in Tables 1 and 2 are marginal.

**Questions:**

see Weaknesses.

---

> ### Author Response · Authors · 2024-11-23
> **Author Response for Reviewer Scc6**
>
> Thank you very much for your suggestions! We sincerely hope our response can help address your concerns. If you have any other questions, we would be more than happy to respond !
>
> > **The paper's efficiency comparison between ControlAR and ControlNet++ (22M vs 361M parameters) is misleading. Comparing parameter counts between diffusion-based and AR-based models is fundamentally unfair due to their different architectures and generation mechanisms. The paper should instead compare inference time, FLOPs, or other more relevant efficiency metrics between similar architectures.**
>
> **A:**
> Thank you for your questions and suggestions.
> We compare the number of additional parameters in the paper to show that our approach does not require a large control encoder to achieve good results, even though the two generative networks are not quite the same in terms of structure and generation mode.
> Additionally, we provide the comparisons about computation burget (MACs, Multiply–Accumulate Operations) in the below table, where both SD1.5 and ControlNet++ set the denoising step to 20. It is clear from the statistical results that the increase in computation of our method is negligible.
>
> |method         |    MACs        |
> |---------------|----------------|
> |SD1.5          |  0.34×20=6.8T  |
> |ControlNet++   |  0.46×20=9.2T  |
> |LlamaGen-XL    |     1.5T       |
> |ControlAR      |  1.5+0.05=1.55T|
>
> Since the control encoder is relatively lightweight, the computational and time overhead introduced by our method is significantly smaller compared to LlamaGen, especially in the generation of high-resolution images, where the additional time cost of ControlAR is negligible.
> These autoregressive image generation models adopt next-token prediction, *e.g.*, LlamaGen directly employs the Llama architecture. However, these methods have not yet fully exploited the efficient inference capabilities of autoregressive models. Some effective acceleration techniques, such as FlashAttention, PagedAttention, and vLLM, are already available for AR models like GPT and Llama3. Therefore, we believe that autoregressive image generation can achieve significant inference speedups. We are also actively exploring commonly used tools, such as vLLM, to enhance the inference speed of ControlAR.
>
>
> > **Q: The use of pretrained visual encoders (CLIP, DINOv2) is a standard practice in multimodal learning**
>
> **A:**  Though CLIP ViT has been widely used for large multi-modal models as the vision encoder,
> our ControlAR introduces DINOv2 as a control encoder for the first time in a controllable image generation task.
> Previous works on controllable generation tend adopt CNNs to encode control images.
> For example, diffusion-based controllable generation task, ControlNet uses half of its own U-net network as the control encoder, and T2I-Adapter designs a simple CNN network as the control encoder. Although these methods have achieved good results in diffusion models, they are not suitable for autoregressive models.
> In this work, we explore the impact of different pre-trained ViTs on controllable image generation, especially for autoregressive models. This is an interesting direction, as different pretraining methods indeed exhibit varying performance.
> Specifically, we compare the effectiveness of ViTs with different pre-training approaches on different data and different tasks, and demonstrate that DINOv2 has better results than other ViTs or CNNs for controllable autoregressive image generation.
>
> > **Q: The quantitative improvements shown in Tables 1 and 2 are marginal.**
>
> Thank you for expressing your opinion on the results of our experiments, but we respectively disagree with it.
> In Tab. 1, we show the quantitative metrics for class-to-image controllable generation of ControlAR based on AiM and LlamaGen. Compared with ControlVAR, our method uses LlamaGen-L with only 343M parameters to achieve lower FID values than VAR-30d with 2B parameters. And when the number of generating model parameters is close, our method has a very clear advantage.
> In Tab. 2, we show the conditional consistency metrics under different control conditions. Our approach has clear advantages compared to ControlNet for different tasks. And compared to ControlNet++, which continues to make targeted fine-tuning of conditional consistency on top of ControlNet, our approach is also very competitive.

---

> ### Author Response · Authors · 2024-11-25
>
> Dear Reviewer,
>
> Hello! May we ask if our response has addressed your concerns? If you have any other questions, we would be more than happy to discuss them with you. We sincerely hope to resolve any doubts you may have and kindly ask you to reconsider our submission. Have a nice day!
>
> Best regards,
> Authors

---

> > ### Comment · Reviewer_Scc6 · 2024-11-29
> >
> > I do not buy the explanation about the novelty of using dinov2, and the performance improvement. Therefore, I will keep my score.

---

> ### Author Response · Authors · 2024-12-01
>
> Thank you very much for your response. I'd like to further elaborate on the two issues you mentioned.
>
> * **Novelty of using DINOv2:**
>
> Firstly, we'd like to clarify that our primary contribution lies in ControlAR, a simple yet effective method for AR-based controllable generation. Moreover, ControlAR is the first controllable generation method based on next-token prediction.
> Compared to traditional diffusion-based controllable generation methods, the AR-based model relies on token-level control encoding. Therefore, we further explore the control encoding method using ViT architectures in this work and evaluate the impact of different pre-training strategies.
>
> Using DINOv2 to encode spatial controls in ControlAR is a novel way compared to previous approaches, such as ControlNet or ControlNet++. Previous control-to-image generation methods rarely explore how to encode spatial controls and overlook existing *off-the-shelf* pre-trained models for controllable generation.
> Compared to other pre-trained ViTs, such as ImageNet, DINOv2 provides superior geometric control encoding. Additionally, due to its pre-training on large-scale data, it demonstrates greater robustness on general datasets.
> In ControlAR, we have highlighted these phenomena and provided substantial evidence. We believe this will offer valuable references and inspiration for future controllable generation models and training strategies.
>
> It is worth emphasizing that *using DINOv2 is not our sole contribution*. Our **primary contribution** lies in ControlAR: an autoregressive controllable generation method. Within this framework, (1) we proposed the DINOv2-based control encoding approach, (2) the conditional decoding method for controllable generation, and the capability for (3) arbitrary-resolution controllable generation.
>
> * **Performance improvement:**
>
> In Tab. 1, we specifically compared the ControlAR method with ControlVAR (a concurrent controllable generation work based on VAR). The results are highly significant:
> * (1) With the same number of parameters (e.g., ~300M), ControlAR achieves a Depth FID of 4.19, while ControlVAR scores 13.8, showing a substantial improvement over ControlVAR.
> * (2) Furthermore, the results from ControlAR (300M parameters) outperform ControlVAR (2B parameters) in terms of FID. This directly demonstrates the superiority of our method—fewer parameters and better performance. Moreover, LLamaGen-L (343M parameters) obtains 3.80 FID on ImageNet while VAR-d30 (2.0B parameters) obtains 1.92 FID on ImageNet, indicating ControlVAR has better generation models while the performance is inferior to our ControlAR.
> We do not consider the results marginal; on the contrary, it is a highly significant improvement!
>
> As for Tab. 2, our ControlAR achieves state-of-the-art results across multiple tasks and significantly outperforms ControlNet overall. These results are not marginal improvements. Notably, ControlNet++ fine-tunes the pre-trained model of ControlNet and employs an additional reward model for supervision, ensuring the generated results are highly consistent with the geometric controls. In contrast, our proposed ControlAR, using a **simple from-scratch** training approach, achieves competitive results with ControlNet++ and outperforms it in some cases. Moreover, it is important to note that the performance of the underlying generative models, such as SD1.5 or LlamaGen, significantly influences the performance of ControlNet and ControlAR models.

---

### Official Review · Reviewer_1dGR · 2024-11-04

**Soundness:** 3
**Presentation:** 3
**Contribution:** 2
**Rating:** 6
**Confidence:** 4

**Summary:**

The paper introduces ControlAR, a method for image-controlled autoregressive-based image generation. A ViT-based encoder is used to extract features from control images such as edge or depth maps, and a conditional decoding method (adding the control features and token features according to a fixed corresponding spatial relationship) is used to generate the output image. This paper has demonstrated good quantative and visualization results. This method also enables image generation at arbitrary resolutions according to the resolution of the control image.

**Strengths:**

- **Important topic**: The image-controlled generation task is in general of great interest, and important for the recently re-rised research trend on AR-based image generation models.
- **Simple and reasonable**: This method is a reasonable exploration towards controlled generation in image AR models with simple token feature addition in decoding.
- **Good ablations** on training strategy, fusion strategy (cross-attention or addition, addition layers), control encoders.
- **Good visualization results**.

**Weaknesses:**

I will put all questions in this section. Note that they're not all weaknesses.

1. **Equation (4) is not written properly**, here $q_i$ represents a discrete image token, then $q_i \in [V]$, where $V$ is the vocabulary or codebook size. Then, the summation of a discrete token with another continuous feature $q_i + C_{i+1}$ in Equation (4) is not well defined. Besides, ControlAR adds up the control feature to the token feature in three intermediate layers, not the input embedding.

2. **About the pathway choice**

    Throughout the paper and in Fig. 2, the authors argue that compared to putting control tokens in the sequence ("conditional prefilling"), ControlAR benefits from a shorter sequence length and eliminates the need for the model to learn a fixed one-to-one spatial mapping. This makes sense that ControlAR is a cheaper way to train a image-controlled AR generation model.

    However, I have some concerns about this path: ControlAR trains each controlled generation task separately, rather than offering a general model. Besides, the predefined one-to-one spatial mapping of ControlAR seems to be restricted to specific tasks. In contrast, putting control tokens in the sequence has the potential to support general generation of texts and images, and will allow flexible and diverse control relationships between them. For example, one might require image-controlled generation based on style rather than spatially local controls like edges or segmap; and one might require referencing multiple control images to generate a single output.

    After all, if the goal is simply to achieve specific, local controls, well-established diffusion models and strategies are already very handy. One of the key motivations of the recent trend in exploring AR image generation models is to achieve a more general and flexible framework that can unify control and generation across modalities, right? I would like to know the authors' opinion on this.

3. **Lack some details and explanations**
    1. About the ablation on control encoders: What are the messages? Is vanilla training better or self-supervised training better for control feature extraction? It's not intuitive that we need different ViT models to extract features from control images for class-to-image and text-to-image generation.
    2. For C2I, the authors initialize the control encoder with VIT-S. The original position encoding is global trainable and of fixed size, and is thus not suitable for multi-resolution. How do the authors handle this?
    3. Section 4.2 lacks detailed information on multi-resolution training. It would be helpful if the authors provided more details, such as the size of the multi-resolution dataset used for training, the design of the architecture (e.g., positional encoding) for multi-resolution adaptation, and so on.
    4. It would be beneficial if the author evaluate the text-image alignment.

4. **About resolution control**

    The paper saied ControlAR extend the ability of the autoregressive model to generate arbitrary resolution, making it "easy" to achieve any-resolution image generation without resolution-aware prompts. I have two questions and suggestions: (1) Can ControlAR or its extension enable resolution control when no specific control image like edge or seg map is available? If so, how? (2) Since both ControlAR and resolution-aware prompts require additional training, it is unclear if ControlAR actually offers a easy solution, despite the intuition tells us so. A quantitative or qualitative comparison with resolution-aware text prompts would strengthen this argument.

Minor: Typo: double "," in the 3rd contribution

**Questions:**

See the weakness section.

---

> ### Author Response · Authors · 2024-11-23
> **Author Response for Reviewer 1dGR [1/3]**
>
> Thank you very much for your suggestions! You have provided many valuable discussions and suggestions. We sincerely hope our response can help resolve your concerns. If you have any other questions or areas of uncertainty, we would be delighted to discuss them with you.
>
>
> > **Q: Equation (4) is not written properly, here $q_i$ represents a discrete image token, then $q_i \in [V]$, where $V$ is the vocabulary or codebook size. Then, the summation of a discrete token with another continuous feature $q_i+C_{i+1}$ in Equation (4) is not well defined. Besides, ControlAR adds up the control feature to the token feature in three intermediate layers, not the input embedding.**
>
> **A:** Thank you for pointing out the inappropriateness of the equation. We modify the formula to the following more intuitive form：
>
> \begin{equation}
>     S_{out}=\mathcal{F}(S_{in} + \mathcal{P}(C)) = \mathcal{F}([c+C_1,I_1+C_2,I_2+C_3,...,I_{i-1}+C_i]),
> \end{equation}
>
> where $\mathcal{F}$ represents a single sequence layer modeling process in the generative network, $\mathcal{P}$ is the projection function, $S_{in}$ and $S_{out}$ are the input sequence and output sequence of each layer respectively, c is the class or text token, $I_i$ is the image token, and $C$ is the control condition sequence.
>
> > **Q: Throughout the paper and in Fig. 2, the authors argue that compared to putting control tokens in the sequence ("conditional prefilling"), ControlAR benefits from a shorter sequence length and eliminates the need for the model to learn a fixed one-to-one spatial mapping. This makes sense that ControlAR is a cheaper way to train a image-controlled AR generation model. However, I have some concerns about this path: ControlAR trains each controlled generation task separately, rather than offering a general model.**
>
> **A:** Thank you for your comments and suggestions. We chose to train ControlAR separately for each control as ControlNet did in order to pursue better control generation, but this does not mean that our ControlAR can't do a general control. In order to prove this, we made additional experimental attempts, we used Dinov2-base as general control encoder to process multiple controls, including canny, hed, lineart and depth. We report evaluation results are shown in the following table. The last line marked with * is the evaluation results of the general model (one model to process different controls). The results show that even as a general model, our ControlAR is competitive to expert models for different controls.
>
> |   Method      | Canny(F1-Score↑)| Hed(SSIM↑)| Lineart(SSIM↑)| Depth(RMSE↓)  |
> |---------------|-----------------|-----------|---------------|---------------|
> | ControlNet    |      34.65      |   76.21   |     70.54     |     35.90     |
> | ControlNet++  |      37.04      |   80.97   |     83.99     |     28.32     |
> | ControlAR     |      37.08      |   85.63   |     79.22     |     29.01     |
> | ControlAR*    |      37.42      |   85.09   |     78.79     |     30.88     |

---

> ### Author Response · Authors · 2024-11-23
> **Author Response for Reviewer 1dGR [2/3]**
>
> > **Q: Besides, the predefined one-to-one spatial mapping of ControlAR seems to be restricted to specific tasks. In contrast, putting control tokens in the sequence has the potential to support general generation of texts and images, and will allow flexible and diverse control relationships between them. For example, one might require image-controlled generation based on style rather than spatially local controls like edges or segmap; and one might require referencing multiple control images to generate a single output. After all, if the goal is simply to achieve specific, local controls, well-established diffusion models and strategies are already very handy. One of the key motivations of the recent trend in exploring AR image generation models is to achieve a more general and flexible framework that can unify control and generation across modalities, right? I would like to know the authors' opinion on this.****
>
> Our ControlAR currently provides an efficient autoregressive model-based technical route for the controllable image generation of spatial structures.
> For controllable image generation based on spatial controls, we believe this one-to-one spatial mapping is a highly efficient and effective approach for autoregressive models. However, we also consider that the **conditional decoding** proposed in this paper is not limited to geometric control generation.
> We believe this control method can be extended to more general controllable generation. Specifically, our control encoder can serve as a universal control encoder, capable of handling geometric controls, content control, or even color and style control.
> Firstly, we encode these diverse controls into a sequence of our expected length (which determines the resolution of the output image), *e.g.*, 512 tokens.
> And then we use the control sequence with **conditional decoding** for the controllable image generation. Compared to prefilling decoding, this approach enables *efficient inference*, *strengthens control over generation*, and supports *arbitrary-resolution generation*.
> For example, this approach can be directly applied to style transfer, where the style image can be encoded into conditional control tokens.
>
> > **Q: About the ablation on control encoders: What are the messages? Is vanilla training better or self-supervised training better for control feature extraction? It's not intuitive that we need different ViT models to extract features from control images for class-to-image and text-to-image generation.**
>
> **A:** We compare the two models ViT-s[1] and DINOv2-s[2] in our ablation experiments on the control encoder. The experimental results show that ViT-s performs better in C2I and DINOv2-s is better for T2I. We believe that the reason for this phenomenon is the different pre-training data for the two models. ViT-s is obtained by pre-training on ImageNet and thus is more advantageous for C2I tasks that are also trained on ImageNet. DINOv2-s, on the other hand, is pre-trained on a larger and more diverse data such as LVD-142M, and thus will be more suitable for T2I tasks trained on MultiGen20M, which is also a diverse text-image paired dataset.
>
> > **Q: For C2I, the authors initialize the control encoder with VIT-S. The original position encoding is global trainable and of fixed size, and is thus not suitable for multi-resolution. How do the authors handle this?**
>
> **A:** When using ViT-s, we interpolate the original positional embeddings based on the input size to adapt to different image resolutions.
>
> > **Q: Section 4.2 lacks detailed information on multi-resolution training. It would be helpful if the authors provided more details, such as the size of the multi-resolution dataset used for training, the design of the architecture (e.g., positional encoding) for multi-resolution adaptation, and so on.**
>
> **A:** In multi-resolution training we first set the maximum sequence length to 2304, supporting a batch size of 2 per A100 GPU under this limit. The control image is downsampled 16 times to obtain the control sequence. For example, when the resolution of the control image is 768×768 or 1024×576, then (768//16)×(768//16)=(1024//16)×(576//16)=2304.
> During the training process we randomly sample the height and width of the training data from 384 to 1024 with a minimum interval of 16, and the image can be resized when it satisfies (H//16)×(W//16)$\leq$2304. In addition, we need to adjust the parameter settings of the rotational position encoding in the generative network by simply increasing its maximum sequence length to 2304.
> We have added the details in the revised version.

---

> ### Author Response · Authors · 2024-11-23
> **Author Response for Reviewer 1dGR [3/3]**
>
> > **Q: It would be beneficial if the author evaluate the text-image alignment.**
>
> **A:** Thanks for your suggestion! We used the **CLIP-score** to evaluate the alignment between the generated images and the text prompts, with experiments conducted on the ADE20K dataset. As shown in the table below, the CLIP-score between text and images in ADE20K is 31.26, primarily due to the use of pseudo-caption annotations, which contain significant noise. Our method achieved a CLIP-score of 30.86, which is very close to the inherent CLIP score of the dataset.
>
> |Method          |   CLIP-score |
> |----------------|--------------|
> |ADE20K          |      31.26   |
> |T2I-Adapter     |      30.65   |
> |UniControlNet   |      30.59   |
> |UniControl      |      30.92   |
> |ControlNet      |      31.53   |
> |ControlNet++    |      31.96   |
> |Ours            |      30.86   |
>
>
> > **Q: About resolution control. The paper saied ControlAR extend the ability of the autoregressive model to generate arbitrary resolution, making it "easy" to achieve any-resolution image generation without resolution-aware prompts. I have two questions and suggestions: (1) Can ControlAR or its extension enable resolution control when no specific control image like edge or seg map is available? If so, how? (2) Since both ControlAR and resolution-aware prompts require additional training, it is unclear if ControlAR actually offers a easy solution, despite the intuition tells us so. A quantitative or qualitative comparison with resolution-aware text prompts would strengthen this argument.**
>
> **A:** Thanks for your questions and suggestions very much!
>
> (1) It's not really difficult to make our ControlAR do things like control resolution even when there's no specific control image input. We can generate a grayscale map of the corresponding resolution according to the desired height and width, this grayscale map consists of the number of 16 × 16 small squares, and the grayscale value of each row decreases from left to right, the leftmost 255, the rightmost 0. This grayscale image is the control image that determines the resolution. Thanks to the strong positional dependence of the control decoding strategy between the image token and the control condition token, the model only needs to generate a sequence as long as the control condition sequence. And since the grayscale value of each row is decreasing from left to right, the model can easily know when it is necessary to switch to the next row. We have verified the feasibility of this approach on a small experimental scale. We provide some visualization results in **Fig. 8 of the revised version**.
>
> (2) Using resolution-aware prompts to control the resolution as in Lumina-mGPT[1] requires the constant generation of `<end-of-line>` tokens during the prediction of the image and the eventual prediction of `<end-of-image>` token. This approach requires the model to make its own decisions about where to make line breaks and where to end generation, but our ControlAR is directly telling the model where to make line breaks and end generation. We try to train our ControlAR using the approach mentioned in (1) and only need to fine-tune the weights based on LlamaGen-XL (512×512) on about 1M text-image paired data for 30k steps to achieve a good arbitrary resolution generation capability without specific control image. This proves that our ControlAR can be a very effective strategy for controlling resolution.
>
> > **Q: Minor: Typo: double "," in the 3rd contribution**
>
> **A:**  Thank you very much for the correction, we have corrected this error in the revised version.
>
>
> References:\
> [1] Dosovitskiy A. An image is worth 16x16 words: Transformers for image recognition at scale[J]. arXiv preprint arXiv:2010.11929, 2020.\
> [2] Oquab M, Darcet T, Moutakanni T, et al. Dinov2: Learning robust visual features without supervision[J]. arXiv preprint arXiv:2304.07193, 2023.\
> [3] https://github.com/huggingface/transformers\
> [4] Liu, Dongyang, et al. "Lumina-mgpt: Illuminate flexible photorealistic text-to-image generation with multimodal generative pretraining." arXiv preprint arXiv:2408.02657 (2024).

---

> > ### Comment · Reviewer_1dGR · 2024-11-24
> > **Response to the authors**
> >
> > Thank you for the detailed response. I particularly like the newly added experiment of handling multiple control signals with one model and the any-resolution generation experiment in Fig. 8. Although I might still have some concerns about my previous question on that *the predefined one-to-one spatial mapping of ControlAR seems to be restricted to specific tasks*, I think the current contributions have already earned this paper an accept. I'll increase my score.
> >
> > Here is my remained concern:
> > > Firstly, we encode these diverse controls into a sequence of our expected length (which determines the resolution of the output image), e.g., 512 tokens. And then we use the control sequence with conditional decoding for the controllable image generation.
> >
> > As far as I understand, your response said that if we want another flexible control, e.g., specified by a prompt, we can use another encoder to encode this control into a sequence of control tokens that is of the sequence length of the image. After this, ControlAR can be applied. If we would like a flexible control, I think this additional encoder needs to be strong. If so, the argued efficiency benefits of ControlAR over a unified method that directly model the joint distribution of interleaved text and images will be reduced.

---

> > > ### Author Response · Authors · 2024-11-24
> > > **Thank you for your recognition!**
> > >
> > > Thank you very much for your recognition and for increasing the score!
> > >
> > > About:
> > > > "Firstly, we encode these diverse controls into a sequence of our expected length (which determines the resolution of the output image), e.g., 512 tokens. And then we use the control sequence with conditional decoding for the controllable image generation."
> > >
> > > I'd like to explain further about how to extend the proposed ControlAR to more general controllable image generation beyond spatial controls, *e.g.,* style transfer, color control or identity-preserving generation. ControlAR can serve as a general paradigm for autoregressive models.
> > >
> > > * *General Control Encoding*: We adopt Control Encoders to encode various control inputs or combined controls (multi-control generation) into control sequences. This modularity enables seamless integration of diverse control inputs, such as:
> > >     - Spatial and geometric controls (e.g., depth maps, segmentation maps).
> > >     - Semantic controls (e.g., text prompts, scene descriptions).
> > >     - Style and appearance controls (e.g., color palettes, artistic styles).
> > >
> > >   We can either share a unified encoder across multiple tasks or use different encoders for different tasks, offering a flexible and adaptable setup. Various control inputs are encoded into a sequence, and we can further control the resolution of the generated image by adjusting the length of the control encoding sequence.
> > >
> > >   For general controllable generation tasks, a ViT model with 22M parameters has already demonstrated excellent performance. The impact of this component on inference efficiency is much smaller compared to the decoding process. If a control encoder needs to handle multiple controls simultaneously, it indeed requires a larger model to ensure sufficient generation quality. However, compared to generative networks like LlamaGen, the parameter size and computational cost of our encoder are significantly smaller. Currently, our inference efficiency is primarily determined by the autoregressive generation network.
> > >
> > >
> > > * *Conditional Decoding*: we use the control sequence with **conditional decoding** to predict the image tokens.
> > >
> > >
> > > If you have further questions, we would be more than happy to discuss and exchange ideas with you. We believe that applying ControlAR to more general controllable generation tasks is an exciting research direction to explore.
> > >
> > > I greatly look forward to and enjoy discussing with you. Have a nice day!

---

### Official Review · Reviewer_uMML · 2024-11-04

**Soundness:** 4
**Presentation:** 4
**Contribution:** 3
**Rating:** 8
**Confidence:** 4

**Summary:**

The paper presents a method to condition an autoregressive generative image model on different modalities, such as edges, depth, and others. The application formulation is very similar to ControlNet for diffusion models, but the method is novel for AR models. The proposed method consists in (1) generating patch embeddings for the conditioning input (2) adding those embeddings to the image embeddings in certain augmented layers in the AR model (3) processing the combined patches normally. The new layers are trained on a large dataset with conditioning inputs and the method achieves strong results.

**Strengths:**

The paper has several strengths that make it compelling:
The work has a very simple formulation that is elegant. There is good demonstration on how it’s better than the other obvious approach of conditional prefilling. Also, very few other work exists tackling this problem and this is, to the best of my knowledge, a novel approach for conditioning AR models. They also present class-to-image and T2I evaluations and show strong results on several datasets.

Also, this direction of research discovers essential knowledge for these new models, which we already have for diffusion models. And we can also see that ControlAR is smaller than a typical ControlNet. Further, the paper has good details on experimental setup, good  ablations, specifically some interesting ones on position+quantity of control layers.

**Weaknesses:**

I don't think I have found weaknesses in the work that should lead to rejection. I am curious about what would happen if certain experiments were run, and these are not very extensive. Some examples:
1. Which layers are ideal to introduce the new control layers on? Right now we have a coarse study of this but it could go deeper, although it's a lot of work that might not be super useful in the end.
2. Some output images shown in the paper show some color saturation or excess contrast - is this an effect of the control layers or just the base model? Is training the control layers biasing the model towards some unrealistic outputs?

**Questions:**

I think I am strongly decided for acceptance given the strengths of the paper. I'll read the other reviews in case I missed anything but currently don't have major questions. The paper presents a straightforward improvement that is necessary for these types of models and does a great job in presenting it, and in evaluating it.

---

> ### Author Response · Authors · 2024-11-23
> **Author Response for Reviewer uMML**
>
> We sincerely appreciate your recognition of our work and genuinely hope that our response addresses your concerns. If you have any further questions, please feel free to let us know!
>
> > **Q: Which layers are ideal to introduce the new control layers on? Right now we have a coarse study of this but it could go deeper, although it's a lot of work that might not be super useful in the end.**
>
> **A:** It's an interesting question! Our ControlAR replace layers **1-th**, **13-th**, and **25-th** of LlamaGen-XL's 36-layer Transformer with the proposed *conditional sequence layer* for adding controls.
> We further analyzethe impact of adding control at different layers based on the **depth-to-image** generation.
>
> |Fusion layer  |  RMSE↓  |  FID↓  |
> |--------------|---------|--------|
> |1,13,25       |  29.01  |  14.61 |
> |13,25         |  30.82  |  16.17 |
> |1,25          |  36.75  |  19.44 |
> |1,13          |  35.74  |  17.21 |
>
> It shows that suppressing the conditional fusion of the middle layer (13-th layer) has the greatest impact on the generated results. It is sincerely hoped that this result will be of some help to you in your research.
>
> > **Q: Some output images shown in the paper show some color saturation or excess contrast - is this an effect of the control layers or just the base model? Is training the control layers biasing the model towards some unrealistic outputs?**
>
> **A:** Thank you for your suggestion! We're inclined to present a visualization that is more visually appealing, so we may have inadvertently selected images with higher contrast. However, our results perform well on the FID metric, which means that our generated images are closer to the real data compared to other methods.

---

> > ### Comment · Reviewer_uMML · 2024-11-25
> >
> > Thank you for the clarifications. These are good answers. I think this paper is very good, I will keep my score.

---

> > > ### Author Response · Authors · 2024-11-26
> > >
> > > Dear Reviewer uMML,
> > >
> > > Thank you very much for recognizing our work! Your suggestions are also extremely valuable! Wishing you a pleasant day!
> > >
> > > Sincerely,\
> > > Authors

---

### Official Review · Reviewer_GBJo · 2024-11-05

**Soundness:** 3
**Presentation:** 3
**Contribution:** 3
**Rating:** 6
**Confidence:** 4

**Summary:**

This paper introduces ControlAR, a method to efficiently enable controllability in autoregressive (AR) image generation models. ControlAR proposes a control encoder that transforms spatial control inputs (e.g., edges, depth maps, segmentation maps) into sequential control tokens, which are leveraged during conditional decoding to enable precise control over the generated images.

**Strengths:**

1. The proposed method enables fine-grained control in autoregressive image generation by using a control encoder and conditional decoding, achieving high image quality with low additional training cost.

2. This method provides effective resolution control, allowing AR models to overcome the limitations of fixed-resolution generation.

**Weaknesses:**

1.  Performance comparisons with recent models such as Lumina-mGPT and Cm3leon (or Anole), such as in segmentation-to-image tasks, would strengthen this paper. Additionally, an analysis or discussion on the potential for integration with these models would be beneficial.

2. Spatial conditions like segmentation maps and Canny edges impose strong constraints on structure diversity in generated outputs. Exploring whether some structural diversity can be incorporated within the conditional decoding step would be beneficial.

3. Need for discussion on representative failure cases. A discussion of representative failure cases among the generated results would provide valuable insights into the limitations of the proposed method and potential areas for improvement.

**Questions:**

Please see the Weaknesses

---

> ### Author Response · Authors · 2024-11-23
> **Author Response for Reviewer GBJo**
>
> We sincerely appreciate your recognition of our work and genuinely hope that our response addresses your concerns. If you have any further questions, please feel free to let us know!
>
> > **Q1: Performance comparisons with recent models such as Lumina-mGPT and Cm3leon (or Anole), such as in segmentation-to-image tasks, would strengthen this paper. Additionally, an analysis or discussion on the potential for integration with these models would be beneficial.**
>
> **A:** Thank you very much for your suggestion, we have added some quantitative comparative results with recent work including OmniGen[1] and Lumina-mGPT[2] , as shown in the following table.
> As for CM3Leon[3] and Anole[4], we attempted to compare CM3Leon (7B parameters) on control-to-image tasks; however, its corresponding code is not publicly available, making a direct comparison impossible. Additionally, Anole (7B parameters) focuses on multimodal autoregressive text-to-image generation and does not explore the control-to-image generation method.
> Additionally, our method does not require any adjustments to the structure of the generative network or modifications to the length of the sequences, which means that we can easily migrate our ControlAR to other autoregressive image generation models, such as Lumina-mGPT.
> Thanks for your suggestion, we have added the comparisons in the revised version.
>
> |  Type     | Method        |   Param.  | Seg(mIoU↑) | Canny(F1score↑) | Hed(SSIM↑) | Depth(RMSE↓) |
> |-----------|---------------|:---------:|:----------:|:--------------:|:----------:|:------------:|
> | Diffusion | ControlNet    |   1.2B    |   32.55    |     34.65       |  76.21     |    35.90     |
> | Diffusion | ControlNet++  |   1.2B    |   43.64    |     37.04       |  80.97     |    28.32     |
> | Diffusion | OmniGen       |   3.8B    |   44.23    |     35.54       |  82.37     |    28.54     |
> |     AR    | Lumina-mGPT   |   7B      |   25.02    |     29.99       |  78.21     |    55.25     |
> |     AR    | ControlAR     |   0.8B    |   39.95    |     37.08       |  85.63     |    29.01     |
>
> > **Q: Spatial conditions like segmentation maps and Canny edges impose strong constraints on structure diversity in generated outputs. Exploring whether some structural diversity can be incorporated within the conditional decoding step would be beneficial.**
>
> **A:** Thank you for your suggestion. This is a good idea! Given the diversity of structures generated, we sometimes do not want the spatial structure of the generated image to be identical to the input control. To achieve this, it is only necessary to skip the operation of fusing the control condition token with the image token with a probability of 50\% when training ControlAR.
> Such an approach ensures ControlAR's generative capability in the absence of control image inputs. At the same time, multiplying the control condition token by a control strength factor $\alpha$ during inference changes the degree of control of the generated result. When $\alpha$ is 1, ControlAR will generate an image exclusively based on the control condition, while when $\alpha$ is 0, the generated results will be related only to the text prompt. Another even simpler way is to freeze the generative network during training, and still adjust the controlled strength by a control strength factor $\alpha$.
> We provide some examples by adjusting the strength factor in **Fig. 7 in the revised version**.
>
> > **Q: Need for discussion on representative failure cases. A discussion of representative failure cases among the generated results would provide valuable insights into the limitations of the proposed method and potential areas for improvement.**
>
> **A:**  Thanks for your advice, we added a discussion on representative failure cases in the revised version. When there is a significant discrepancy between the text prompts and the spatial controls, ControlAR may produce some results that are not consistent with the text prompts. For more details, please refer to **Fig. 9 in the revised version**.
>
> References:\
> [1] Xiao, Shitao, et al. "Omnigen: Unified image generation." arXiv preprint arXiv:2409.11340 (2024).\
> [2] Liu, Dongyang, et al. "Lumina-mGPT: Illuminate flexible photorealistic text-to-image generation with multimodal generative pretraining." arXiv preprint arXiv:2408.02657 (2024).\
> [3] Yu, Lili, et al. "Scaling autoregressive multi-modal models: Pretraining and instruction tuning." arXiv preprint arXiv:2309.02591 2.3 (2023).\
> [4] Chern, Ethan, et al. "Anole: An open, autoregressive, native large multimodal models for interleaved image-text generation." arXiv preprint arXiv:2407.06135 (2024).

---

> > ### Comment · Reviewer_GBJo · 2024-11-26
> > **Remaining concern about the structural diversity**
> >
> > Thank you for the detailed clarification.
> >
> > While the authors provide a detailed explanation, a concern about the structural diversity remains. Specifically, in the additional experiments provided in Fig. 7, it appears that even when the value of the control strength factor $\alpha$ is varied, the spatial structure conditioned by Canny edges remains strongly preserved, which brings us back to the initial question.
> >
> > This observation aligns with the discussed failure cases shown in Fig. 8, where strong constraints from the conditioning input prevent ControlAR from incorporating structural diversity, even when the text prompt specifies features like "burning candle" or "glasses."
> >
> > In practice, it is unlikely that the Canny edges used as conditioning inputs will perfectly match the user's intended structure. Instead, they are more often provided as approximate guidance, with the expectation that the text prompt will introduce additional variations or guide the generation toward more diverse interpretations.
> >
> > Given this, it seems that ControlAR struggles to relax such structural constraints, thereby limiting the structural diversity of the generated outputs. Do you think this could be considered a noteworthy limitation of ControlAR? If not, could you provide further evidence or results to demonstrate ControlAR’s ability to generate diverse structures under such conditions?

---

> ### Author Response · Authors · 2024-11-25
>
> Dear Reviewer,
>
> Hi! May we ask if our response has addressed your concerns? If you have any other questions, we would be more than happy to discuss them with you. In the revision version, we have added many new experimental results that can help address your concerns.
>
> Best regards,
> Authors

---

> ### Author Response · Authors · 2024-11-27
> **Response to Reviewer GBJo about structural diversity**
>
> Dear Reviewer,
>
> Thank you very much for your response!
>
> You have raised a very meaningful question. In response, we have further **revised the submission, adding additional results and comparisons**. We hope that our reply can address your concerns.
>
> In the examples previously shown in Fig. 7, we can observe that as the alpha value changes, the influence of canny control on the generated images gradually decreases. To further demonstrate the role of $\alpha$ in geometric control, we have added more visualization results of control coefficients to Fig. 7 (**the updated revision**), including Canny edges, HED edges, and LineART controls. Fig. 7 illustrates that as the control $\alpha$  decreases, the differences between the generated results and the spatial controls become increasingly significant. Additionally, this results in different image layouts and the diversity of geometric structures is also improved.
> Therefore, we believe that ControlAR with $\alpha$ allows for **generating images that are both aligned with spatial controls and exhibit structural diversity**.
>
> Regarding the second issue, we must admit that **all current control-to-image models face this challenge: the conflict between text prompts and geometric controls**. This issue is prevalent in control-to-image models such as *ControlNet* and *ControlNet++* (as shown in Fig. 9). These well-established diffusion models struggle to balance the text prompts and spatial controls. However, we believe that these control-to-image models are currently focused on generating results that align with spatial controls. In fact, ControlNet++ introduces additional supervision to promote alignment between the generated image and spatial controls, which weakens the influence of the text prompt. Therefore, in the context of controllable generation, these control-to-image models, including our proposed ControlAR, will all try their best to generate results that align with the spatial controls.
>
> However, in ControlAR, we explore a dynamic way of adjusting spatial control, which allows ControlAR to reduce its adherence to spatial controls and generate results with more structural diversity, as shown in Fig. 7. Similarly when facing the conflict between text prompts and spatial controls, ControlAR can mitigate these conflicts by adjusting the coefficient $\alpha$ of the spatial controls, enabling the generated results to balance both the text and the control. As shown in Fig. 9, when the coefficient $\alpha$ is set to 0.4, ControlAR can generate elements such as "candles" and "cake," which appear in the text prompt.
>
> Even though current mainstream control-to-image methods encounter similar issues, the ControlAR we propose shows great potential in handling these conflicts effectively.
>
>
> We hope that our response and the updated revision can address your concerns. If you have any further questions, we would be more than happy to discuss them with you. If our response resolves your concerns, we would also appreciate the possibility of an increased score.
>
>
> Sincerely,\
> Authors

---

> > ### Comment · Reviewer_GBJo · 2024-11-29
> >
> > Thank you for the additional results. However, it appears that ControlAR still demonstrates limited ability to effectively control structural diversity, and the accompanying analysis remains insufficient.
> >
> > Nonetheless, considering that this work represents an early attempt to integrate spatial controls into AR models, I find the reasons to accept outweigh the reasons to reject and will maintain my score.

---

> ### Author Response · Authors · 2024-12-02
>
> Dear Reviewer GBJo,
>
> Thank you very much for your reply!
>
> **Conditional consistency** is very important among the evaluation criteria for controllable generation of spatial structures, and existing methods based on diffusion models tend to enforce the generated images to have spatial structures that are as similar as possible to the input conditional images.
>
> I believe the **structural diversity** you mentioned is a thought-provoking issue and a valuable direction for future exploration. However, we'd like to clarify that control consistency and geometric structural diversity are inherently contradictory. If control consistency is high, the structure will inevitably lack diversity; conversely, if structural diversity is high, control consistency will significantly decrease.
> As a result, controllable image generation methods generally struggle to achieve geometric diversity, which we believe is a **common limitation across many approaches, such as ControlNet++ / ControlNet (see Fig. 9), which focus on improving consistency with spatial controls**.  However, inspired by your insights, our ControlAR provides a simple yet effective way to mitigate this issue.
> By adjusting the control strength factor properly, the generated image can take into account the spatial structure and text prompt.
>
> The issue you raised is indeed very meaningful for the future of controllable image generation. Perhaps we should consider building a more multidimensional and comprehensive evaluation framework for controllable image generation, which not only takes into account the **consistency of the controllable generation** but also evaluates whether it can produce **more diverse structures based on spatial controls**. I believe this is an excellent question to explore.
>
> Best regards,\
> Authors

---

### Meta-Review · Area_Chair_5RrA · 2024-12-24

**Metareview:**

This paper introduces ControlAR, a framework to integrate spatial controls into autoregressive image generation models. It enables AR-based controllable image generation by introducing a lightweight control encoder and a conditional decoding strategy. This approach generates each token by fusing control tokens with image tokens, enhancing both efficiency and controllability. The framework supports multiple control modalities and enables arbitrary-resolution image generation. Experimental results demonstrate strong performance across a range of tasks, including edge, depth, and segmentation-based generation, competing with state-of-the-art methods such as ControlNet++.

Overall, this paper makes a significant contribution to the field of controllable image generation. While some limitations remain, the novelty of the approach and the robustness of the experimental results clearly outweigh these concerns. The majority of the reviewers also holds positive feedback, and thus I recommend accepting the paper.

**Additional Comments On Reviewer Discussion:**

The reviewers raised a few questions regarding structural diversity, efficiency metrics, novelty, and other aspects. However, these questions do not represent fundamental flaws in the paper, and the authors have adequately addressed most of the concerns. The significance of this work, as one of the first attempts to enable controllability in autoregressive visual generative models, outweighs these issues. Therefore, I recommend accepting the paper.

---

### Decision · Program_Chairs · 2025-01-22

Accept (Poster)